# Identification of poly(ADP-ribose) polymerase 9 (PARP9) as a noncanonical sensor for RNA virus in dendritic cells

Junji Xing [1], Ao Zhang[1,2], Yong Du[1], Mingli Fang[1,3], Laurie J. Minze[1], Yong-Jun Liu[4], Xian Chang Li[1,5] & Zhiqiang Zhang [1,5 ✉]

Innate immune cells are critical in protective immunity against viral infections, involved in sensing foreign viral nucleic acids. Here we report that the poly(ADP-ribose) polymerase 9 (PARP9), a member of PARP family, serves as a non-canonical sensor for RNA virus to initiate and amplify type I interferon (IFN) production. We find knockdown or deletion of PARP9 in human or mouse dendritic cells and macrophages inhibits type I IFN production in response to double strand RNA stimulation or RNA virus infection. Furthermore, mice deficient for PARP9 show enhanced susceptibility to infections with RNA viruses because of the impaired type I IFN production. Mechanistically, we show that PARP9 recognizes and binds viral RNA, with resultant recruitment and activation of the phosphoinositide 3-kinase (PI3K) and AKT3 pathway, independent of mitochondrial antiviral-signaling (MAVS). PI3K/AKT3 then activates the IRF3 and IRF7 by phosphorylating IRF3 at Ser385 and IRF7 at Ser437/438 mediating type I IFN production. Together, we reveal a critical role for PARP9 as a non-canonical RNA sensor that depends on the PI3K/AKT3 pathway to produce type I IFN. These findings may have important clinical implications in controlling viral infections and viral-induced diseases by targeting PARP9.

[1] Department of Surgery and Immunobiology and Transplant Science Center, Houston Methodist, Houston, TX, USA. [2] Department of Laboratory Medicine, State Key Laboratory of Oncology in South China, Collaborative Innovation Center for Cancer Medicine, Sun Yat-sen University Cancer Center, Guangzhou, China. [3] Department of Molecular Biology, College of Basic Medical Sciences, Jilin University, Changchun, China. [4] Innovent Biologics Inc, Suzhou, China. [5] Department of Surgery, Weill Cornell Medical College, Cornell University, New York, NY, USA. ✉email: zzhang@houstonmethodist.org

Innate immune cells are the first line of defense against invading pathogens including viruses through multiple mechanisms. One of the key defense mechanisms involves the pattern recognition receptors (PRRs) that monitor the cytosol for viral pathogen-associated molecular patterns (PAMPs), such as atypical viral nucleic acids. After recognizing the PAMPs, PRRs are activated and signal to the production of interferon (IFN) and other antiviral or proinflammatory mediators[1]. Type I IFN, including IFN-α and IFN-β, elicit host innate immune response against viral infections. For sensing the virus-derived nucleic acids in the cytosol, many PRRs have been identified, such as viral RNA sensors RIG-I like receptors (RLRs), viral DNA sensors cGAS (cyclic GMP-AMP synthase), IFI16 (IFN gamma-inducible protein 16), DAI (DNA-dependent activator of IFN-regulatory factors), DDX41 and many others[2].

RNA viruses such as severe acute respiratory syndrome (SARS) coronavirus, influenza virus, Zika virus, and human immunodeficiency virus are major threats to human health, because they can cause severe diseases whenever there is an outbreak in humans. The 2019-20 Wuhan coronavirus outbreak is example of ongoing global challenges now caused by RNA virus SARS coronavirus 2[3], which demands a fundamental understanding of the mechanisms of viral pathogenicity and antiviral immunity. In general, RNA viruses can be sensed by PRRs in the cytosol in different cell types. RLRs including RIG-I (retinoic acid-inducible gene I) and MDA5 (melanoma differentiation factor 5), are the major RNA sensors during RNA viral infections; they belong to the DExD/H-box family of helicases[4–6]. RLRs signaling induces MAVS (mitochondrial antiviral signaling) (also known as IPS-1, CARDIF, or VISA) activation and oligomerization into a prion-like aggregate, which activates the TBK1 and IKK kinases. This culminates in the activation of transcription factors NF-κB, IRF3, and IRF7, which translocate to the nucleus to induce type I IFN and other antiviral genes[7]. Other helicases, including DDX60[8], DHX9[9], DDX3[10], DHX15[11], DDX29[12], DHX33[13,14], the DDX1-DDX21-DHX36 complex[15], HMGB proteins[16] and NOD2[17], are also reported as sensors of cytosolic RNA. Most RNA sensors trigger IFN transcription using RLR-dependent pathways (through RLRs themselves or MAVS)[18–20]. However, there remains a major knowledge gap in our understanding of how these receptors recognize and physically bind viral nucleic acids, and whether additional receptors or co-receptors exist. Notably, whether other cytosolic RNA sensors in innate immune cells can trigger MAVS-independent type I IFN production pathway remains largely unknown.

Human PARP family consists of 17 members, which are expressed in a wide variety of tissues and cell types, some ubiquitously expressed while others are in a more restricted manner[21]. The field has evolved from simply focusing on the role of PARPs in DNA damage repair[22] to a diversity of biological responses including cell survival and cell death[23], chromatin structure regulation[24], RNA biology[25,26] and antiviral responses[27]. PARP9 is an inactive mono-ADP-ribosyltransferase in PARP family and has been shown to play some roles in a series of solid tumors[28–30] and macrophages regulation[31] and antiviral immunity[32,33]. Given the multifaceted functions of PARP family proteins in various cellular processes, we carried out a systemic screen of 17 PARP family members in the activation of IFN and NF-κB, hallmarks of innate immune signaling pathways. Here we found that PARP9 was highly induced by IFN-α in dendritic cells (DCs) and played an essential role in producing type I IFN by serving as a MAVS-independent RNA sensor during RNA virus infection. We demonstrated that PARP9 preferentially recognized and bound viral dsRNA of 1.1 kb to 1.4 kb, and then employed phosphoinositide 3-kinase (PI3K) and AKT3 pathway to phosphorylate and activate IRF3 at Ser385 and IRF7 at Ser437/438 for type I IFN production. Thus, our study identified PARP9 as a non-canonical RNA sensor for RNA viruses, and delineated an unknown function for PARP family proteins in antiviral immunity in DCs.

## Results

**PARP9 mediates type I IFN production in human innate immune cells.** Poly (ADP-ribose) polymerase (PARP) family proteins are commonly involved in DNA repair, genomic stability, and programmed cell death[21]. Considering the multifaceted roles of PARP family proteins in vivo, we screened all 17 members of the PARP family proteins in human THP-1 macrophages by a small-interfering-RNA approach. We found that PARP9 knockdown in human THP-1 macrophages led to markedly reduced (over threefold) production of IFN-α in response to cytosolic dsRNA long poly I:C (LPIC, high molecular weight poly I:C), the ligand of RLRs, when delivered via Lipofectamine 3000 (Supplementary Fig. 1a). However, PARP1 knockdown in THP-1 macrophages resulted in decreased IL-6, but not IFN-α in response to LPIC (Supplementary Fig. 1b). We next knocked down PARP9 via short hairpin RNA (shRNA) in THP-1 macrophages. Two distinct PARP9-targeting shRNAs produced efficient knockdown of PARP9 at both mRNA and protein levels (Fig. 1a and b). We then stimulated these cells with LPIC and measured type I IFN and cytokine IL-6 by enzyme-linked immunosorbent assay (ELISA). As a positive control, knockdown of the key adapter MAVS in RLRs signaling pathway via shRNA abrogated the production of type I IFN (IFN-α and IFN-β) and IL-6 in THP-1 macrophages stimulated by cytosolic LPIC (Fig. 1c–e). Similarly, knockdown of PARP9 markedly reduced production of type I IFN, but not IL-6, by THP-1 macrophages compared to cells treated with control shRNA (sh-Ctrl) (Fig. 1c–e). To determine the role of PARP9 in primary cells, we first analyzed expression of PARP9 in human primary cells by microarray gene expression analysis[34,35]. PARP9 had very low expression in most of cell types, but was highly induced by RNA virus influenza A virus (Flu A) and IFN-α in human plasmacytoid dendritic cells (pDC) and myeloid dendritic cells (mDC) (Fig. 1f). Indeed, PARP9 was one of the most highly induced PARP family proteins in pDC after influenza A virus infection (Supplementary Fig. 1c). Furthermore, immunoblot analysis further confirmed that PARP9 was highly induced by IFN-α in human primary pDC and mDC from one random donor (Fig. 1g). To further investigate if PARP9 mediated type I IFN production in dendritic cells, we knocked down either PARP9 or MAVS in human monocyte-derived dendritic cells (MDDC) by shRNA against PARP9 or MAVS (Fig. 1h and i) and stimulated those cells with cytosolic LPIC, dsDNA from HSV-1 virus (HSV-60), cGAMP (STING stimulator in DNA sensing pathway), or dsRNA virus reovirus (Reo) infection for detecting type I IFN production. Knockdown of either PARP9 or MAVS reduced the production of IFN-α (Fig. 1j) and IFN-β (Fig. 1k) in human MDDC in response to LPIC or reovirus infection, but not to dsDNA HSV-60 or cGAMP treatment (Fig. 1j and k). The expression of PARP9 was significantly induced at highest level in human MDDC at 16 h after IFN-α treatment and then dropped at 24 h (Fig. 1l). The kinetics of LPIC-induced IFN-β showed IFN-β reached highest level in control (sh-Ctrl) and PARP9 knockdown human MDDC at 24 h after LPIC stimulation and the difference was the most significant (Fig. 1m). Taken together, these data suggested that human PARP9 mediated the type I IFN production in human innate immune cells including macrophages and DCs in response to dsRNA stimulation and RNA virus infection.

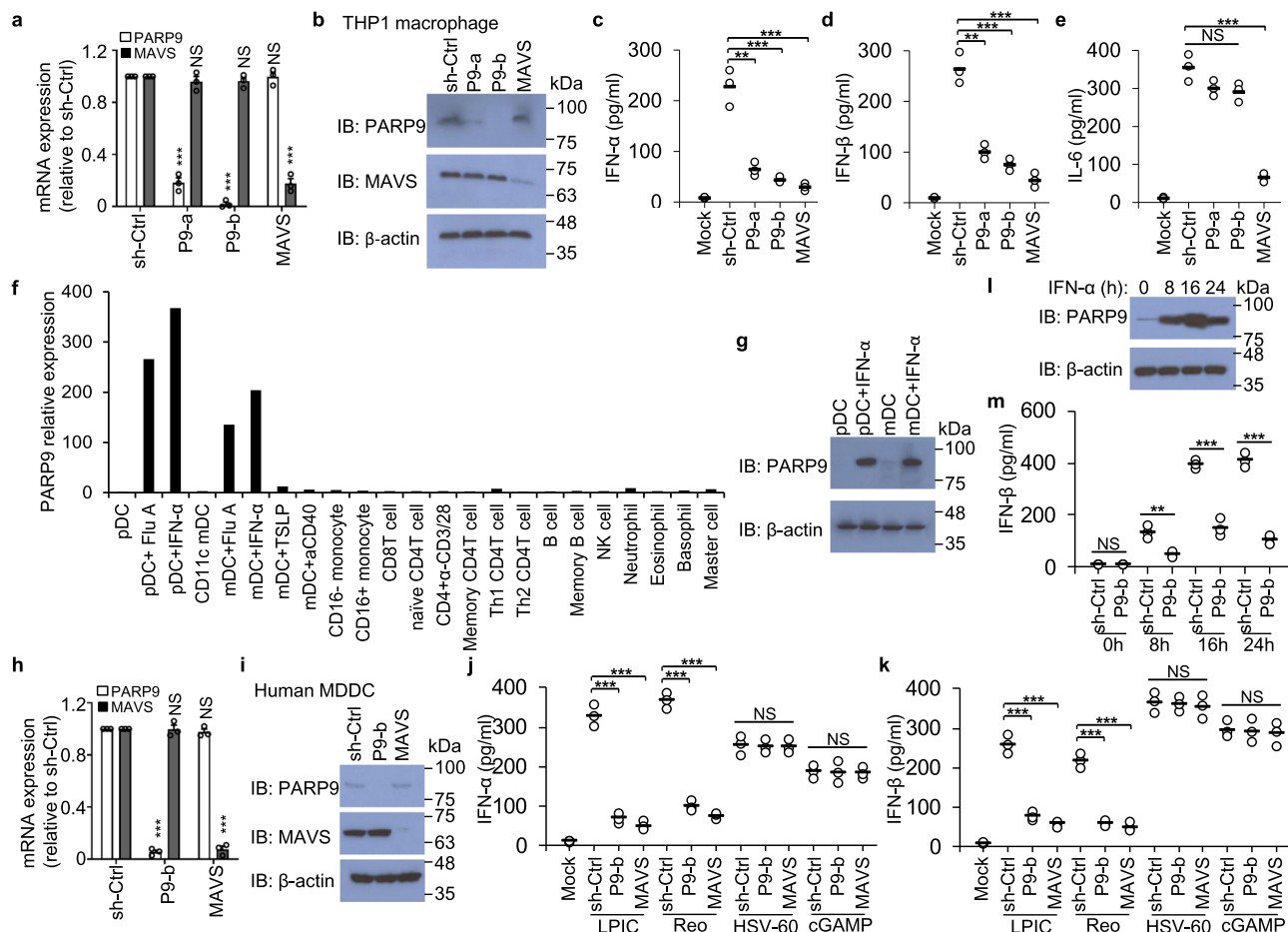

**Fig. 1 PARP9 promotes type I IFN production in human innate immune cells in response to poly I:C stimulation or RNA viral infection.** The RT-qPCR (**a**) and immunoblot (IB, **b**) analysis of PARP9 or MAVS at mRNA (**a**, $n = 3$ per group) and protein (**b**) levels in THP-1 macrophages treated with shRNA to knockdown expression of PARP9 (sequence P9-a and P9-b) or MAVS. A scrambled shRNA served as a control (sh-Ctrl). The β-actin served as the loading control. ELISA of IFN-α (**c**) IFN-β (**d**), and IL-6 (**e**) production from THP-1 macrophages treated with the indicated shRNA after a 10 h stimulation with 0.5 μg/ml of long poly I:C delivered by Lipofectamine 3000 ($n = 3$ per group). **f** Human myeloid cells and lymphoid cells were purified from PBMCs using a cell sorter. Total RNA was isolated from these primary cells induced or not to chip hybridization and microarray. The profile of PARP9 expression in different cells is indicated. The relative expression of PARP9 was compared by plotting the values extracted from the gene expression database. A value <1 indicated the absence of gene expression. **g** Immunoblot (IB) analysis of PARP9 and β-actin in plasmacytoid dendritic cells (pDCs), myeloid dendritic cells (mDCs), and pDCs and mDCs induced with IFN-α (100 U/ml) for 2 h. The RT-qPCR (**h**, $n = 3$ per group) and immunoblot (**i**) analysis of PARP9 or MAVS at mRNA (**h**) and protein (**i**) levels in human monocyte-derived dendritic cells (MDDC) treated with shRNA to knockdown expression of PARP9 (sequence P9-b) or MAVS. A scrambled shRNA served as a control (sh-Ctrl). ELISA of IFN-α (**j**) and IFN-β (**k**) production from human MDDC treated with the indicated shRNA after a 10 h stimulation with 0.5 μg/ml of long poly I:C (LPIC), dsDNA from HSV-1 virus (HSV-60, 2.5 μg/ml) or cGAMP (1.0 μg/ml) delivered by Lipofectamine 3000 or 12 h infection with Reovirus (Reo) at an MOI of 5 ($n = 3$ per group). **l** Immunoblot analysis of PARP9 expression dynamics in human MDDC treated with IFN-α (500 U/ml) for 0 h, 8 h, 16 h and 24 h. **m** ELISA of IFN-β production dynamics in human MDDC treated with the indicated shRNA after stimulation with 0.5 μg/ml of LPIC for indicated time ($n = 3$ per group). Each circle represents an individual independent experiment and small solid black lines indicate the average of triplicates for results in **c**–**e**, **j**, **k** and **m**. Mock, scrambled shRNA-treated cells without stimulation. Error bars indicate standard error of the mean for results in **a**, **h**. NS, not significant ($p > 0.05$), $*p < 0.05$, $**p < 0.01$, $***p < 0.001$, and $p$ value was calculated by unpaired two-tailed Student's $t$ test. Data are from one experiment with duplicate (**f**) or representative of three independent experiments (**a**–**e**, **j**–**m**). Exact $p$ values (**a**, $p = 0.000026$, $p = 0.378$, $p < 0.000001$, $p = 0.276$, $p = 0.929$, $p = 0.000028$; **c** $p = 0.0019$, $p = 0.00099$, $p = 0.00077$; **d** $p = 0.0011$, $p = 0.00057$, $p = 0.00035$; **e**, $p = 0.063$, $p = 0.00018$; **h** $p < 0.000001$, $p = 0.924$, $p = 0.346$, $p < 0.000001$; **j** $p = 0.00014$, $p = 0.000095$, $p = 0.000053$, $p = 0.000024$, $p = 0.826$, $p = 0.73$; **k** $p = 0.00025$, $p = 0.00013$, $p = 0.00018$, $p = 0.00019$, $p = 0.65$, $p = 0.65$; **m** $p = 0.492$, $p = 0.0066$, $p = 0.00033$, $p = 0.00009$).

## The essential role of PARP9 in sensing RNA virus infection in vitro.

To further determine the role of PARP9 in innate immunity in mice, we generated PARP9 knockout (KO) mice and the PARP9 deletion was confirmed by PCR assisted genotyping and immunoblot analysis (Supplementary Fig. S2a–d). We first prepared mouse innate immune cells including bone marrow-derived dendritic cells (BMDC) and bone marrow-derived macrophages (BMDM) from wild-type (WT) and PARP9 knockout (KO) mice, and stimulated those cells with poly I:C (the ligands for RLRs) including LPIC and short poly I:C (SPIC, low molecular weight poly I:C), 5′-triphosphate RNA (5′pppRNA or 5′ppp), dsDNA HSV-60 or cGAMP. ELISA data showed that PARP9 KO BMDC produced much less type I IFN than WT BMDC in response to LPIC, SPIC, or 5′pppRNA stimulation (Fig. 2a and b). Similarly, compared with WT BMDM, there was much less type I IFN in PARP9 KO BMDM in response to poly I:C or 5′pppRNA stimulation (Fig. 2c and d). These data indicated an essential role of PARP9 for type I IFN production in BMDC

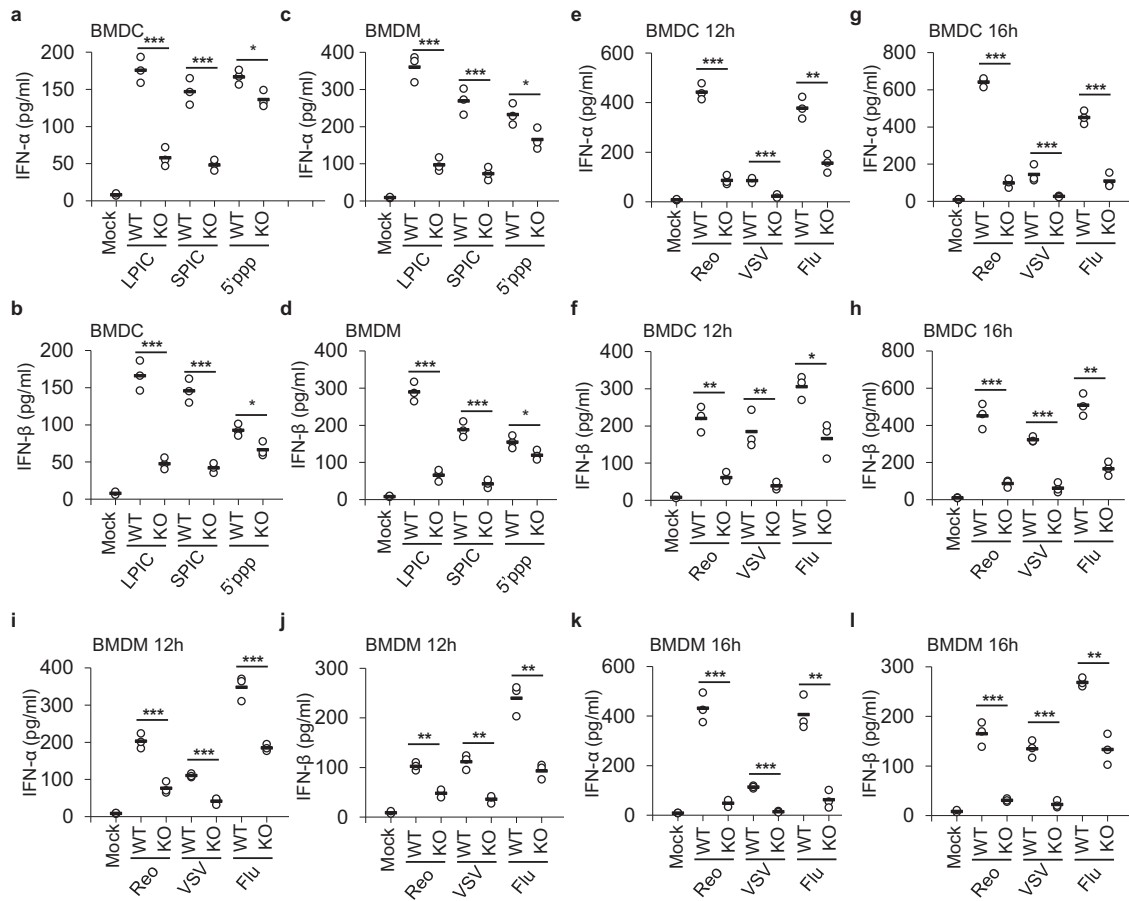

**Fig. 2 PARP9 plays an essential role in type I IFN production in response to infection by RNA viruses.** ELISA of IFN-α (**a**, **c**) and IFN-β (**b**, **d**) production by BMDC (**a**, **b**) or BMDM (**c**, **d**) from wild-type (WT) and *Parp9⁻/⁻* (KO) mice after 10 h of stimulation with long poly I:C (LPIC, 0.5 μg/ml), short poly I:C (SPIC, 0.5 μg/ml) and 5′pppRNA (5′ppp, 0.5 μg/ml) delivered by Lipofectamine 3000 (*n* = 3 per group). ELISA of IFN-α (**e**, **g**) and IFN-β (**f**, **h**) production by BMDC from wild-type (WT) and *Parp9⁻/⁻* (KO) mice after 12 h (**e**, **f**) or 16 h (**g**, **h**) of infection with Reovirus (Reovirus type 3 strain dearing T3D, Reo), VSV (Vesicular stomatitis virus Indiana strain, VSV) or influenza A virus (influenza A virus PR8 strain, Flu) (*n* = 3 per group). ELISA of IFN-α (**i**, **k**) and IFN-β (**j**, **l**) production by BMDM from wild-type (WT) and *Parp9⁻/⁻* (KO) mice after 12 h (**i**, **j**) or 16 h (**k**, **l**) of infection with Reovirus, VSV or Flu virus at an MOI of 5 (*n* = 3 per group). Each circle represents an individual independent experiment and small solid black lines indicate the average of triplicates for results in **a–l**. *$p < 0.05$, **$p < 0.01$, ***$p < 0.001$, and *p* value was calculated by unpaired two-tailed Student's *t* test. Mock, wild-type BMDC or BMDM without stimulation or infection. Data are representative of three independent experiments. Exact *p* values (**a**, $p = 0.00072$, $p = 0.00088$, $p = 0.112$; **b**, $p = 0.00069$, $p = 0.00048$, $p = 0.0495$; **c**, $p = 0.00036$, $p = 0.00098$, $p = 0.0495$; **d**, $p = 0.00023$, $p = 0.00044$, $p = 0.0495$; **e**, $p = 0.00008$, $p = 0.00079$, $p = 0.0025$; **f**, $p = 0.0017$, $p = 0.0081$, $p = 0.0136$; **g**, $p = 0.00001$, $p = 0.00098$, $p = 0.00036$; **h**, $p = 0.00089$, $p = 0.00013$, $p = 0.0012$; **i**, $p = 0.00036$, $p = 0.00002$, $p = 0.00004$; **j**, $p = 0.0012$, $p = 0.0015$, $p = 0.002$; **k**, $p = 0.00043$, $p = 0.00001$, $p = 0.0019$; **l**, $p = 0.00075$, $p = 0.00052$, $p = 0.002$).

and BMDM in response to RLRs ligand poly I:C. However, the levels of proinflammatory cytokines (IL-6 and TNF-α) produced by WT and PARP9 KO BMDC (Supplementary Fig. S3a and b) or BMDM (Supplementary Fig. S3c and d) were statistically no difference in response to poly I:C or 5′pppRNA stimulation (Supplementary Fig. S3a–d), suggesting that PARP9 plays no role in producing proinflammatory cytokines in response to RLRs ligands in BMDC and BMDM. Similarly, BMDC (Supplementary Fig. S3e) or BMDM (Supplementary Fig. S3f) from WT and PARP9 KO mice were comparable in terms of IFN-β production in response to LPS, dsDNA HSV-60, or cGAMP stimulation.

During infection with RNA viruses, the viral genome RNA generates 5′-triphosphate RNA and/or dsRNA in ample amounts during viral replication. Generally, influenza virus or VSV infection produce typical 5′-triphosphate RNA and/or dsRNA, whereas reovirus infection produces dsRNA[36]. Therefore, we employed these RNA viruses including reovirus (T3D strain, Reo), VSV (Indiana strain), and influenza A virus (PR8 strain, Flu) to investigate PARP9 in antiviral innate immunity. BMDC

and BMDM from WT and PARP9 KO mice were isolated and infected with RNA viruses including reovirus, VSV or Flu virus. Compared with WT BMDC, PARP9 KO BMDC produced two- to sixfold less type I IFN at 12 h (Fig. 2e and f) or 16 h (Fig. 2g and h) post-infection by reovirus, VSV or Flu virus. Similar results were also observed at 8 h post-infection by these RNA viruses (Supplementary Fig. S4a and b). Similarly, PARP9 KO BMDM produced two- to fivefold less type I IFN than WT BMDM at 8 h (Supplementary Fig. S4c and d), 12 h (Fig. 2i and j) or 16 h (Fig. 2k and l) post-infection by those RNA viruses. However, the BMDC (Supplementary Fig. S5a–c) or BMDM (Supplementary Fig. S5d–f) from WT and PARP9 KO mice were comparable in terms of the proinflammatory cytokine IL-6 after infection by those RNA viruses.

Reconstitution experiments are a crucial control for knockout specificity. We investigated if overexpression of PARP9 could rescue the phenotype of PARP9 KO BMDC and BMDM. PARP9 was overexpressed in PARP9 KO BMDC (Supplementary Fig. S6a) or BMDM (Supplementary Fig. S6b) by transducing

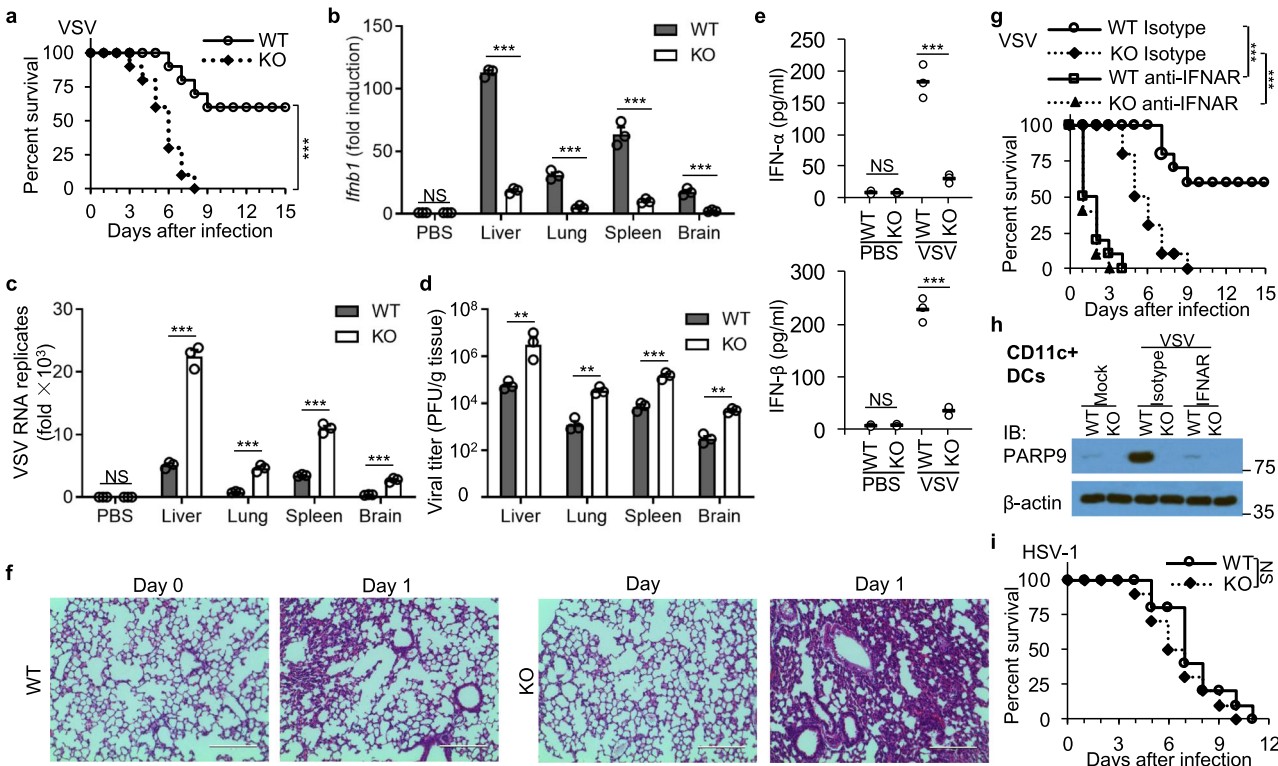

**Fig. 3 PARP9 plays a critical role in host defense against RNA virus infection in vivo. a** Survival of age- and sex-matched PARP9$^{+/+}$ (WT) and PARP9$^{-/-}$ (KO) mice after intraperitoneal infection with VSV ($5 \times 10^8$ PFU per mouse) ($n = 10$ per group). **b** The qRT-PCR analysis of Ifnb1 mRNA in the liver, lung, spleen, and brain of PARP9$^{+/+}$ (WT) and PARP9$^{-/-}$ (KO) mice (three per group) infected for 24 h by intraperitoneal injection of phosphate-buffered saline (PBS) or VSV ($5 \times 10^7$ PFU per mouse); results are presented relative to those of untreated wild-type cells ($n = 3$ per group). **c** The qRT-PCR analysis of VSV RNA in the liver, lung, spleen, and brain of mice as in **b**; results are presented as in **b** ($n = 3$ per group). **d** Plaque assay of VSV titers in the liver, lung, spleen, and brain of mice as in **b** ($n = 3$ per group). **e** ELISA of IFN-β and IFN-α in sera from mice as in **b** ($n = 3$ per group). Each circle represents an individual independent experiment and small solid black lines indicate the average of triplicates. **f** Hematoxylin and eosin (H&E)-staining of lung sections from mice as in **b**. Scale bars represent 200 μm. **g** Survival of age- and sex-matched PARP9$^{+/+}$ (WT) and PARP9$^{-/-}$ (KO) mice after intraperitoneal infection with VSV ($5 \times 10^8$ PFU per mouse) treated with 500 μg anti-IFNAR antibody or isotype control antibody starting 1 day before infection, followed by 250 μg three times per week for up to 2 weeks ($n = 10$ per group). **h** Immunoblot analysis of PARP9 expression in mouse CD11c+ splenic DCs from WT and PARP9 KO mice treated as in **g** for 1 day. **i** Survival of age- and sex-matched PARP9$^{+/+}$ (WT) and PARP9$^{-/-}$ (KO) mice after intravenous infection with HSV-1 ($1 \times 10^7$ PFU per mouse) ($n = 10$ per group). Error bars indicate standard error of the mean for results in **c**, **d**. NS, not significant ($p > 0.05$), $**p < 0.01$, $***p < 0.001$, $****p < 0.0001$, and $p$ value was calculated by unpaired two-tailed Student's $t$ test and Gehan–Breslow–Wilcoxon test for survival analysis. Data are representative of three independent experiments. Exact $p$ values (**a**, $p = 0.0004$; **b**, $p = 1$, $p < 0.00001$, $p = 0.00034$, $p = 0.00068$, $p = 0.00082$; **c** $p = 1$, $p = 0.00009$, $p = 0.00007$, $p = 0.00028$, $p = 0.00028$; **d**, $p = 0.0082$, $p = 0.0014$, $p = 0.00064$, $p = 0.0013$; **e** upper, $p = 0.84$, $p = 0.00063$, lower, $p = 0.85$, $p = 0.00017$; **g** $p < 0.0001$, $p < 0.0001$; **i**, $p = 0.1776$).

with a lentiviral vector expressing wild-type PARP9. As expected, overexpression of PARP9 in PARP9 KO BMDC (Supplementary Fig. S6c and d) or BMDM (Supplementary Fig. S6e and f) produced significantly more IFN-α (Supplementary Fig. 6c and e) and IFN-β (Supplementary Fig. 6d and f) than those cells expressing control vector in response to poly I:C. Similarly, compared with those cells expressing control vector, there were dramatically more IFN-α (Supplementary Fig. S6g and i) and IFN-β (Supplementary Fig. S6h and j) production by PARP9 KO BMDC (Supplementary Fig. S6g and h) or BMDM (Supplementary Fig. 6i and j) overexpressing PARP9 after infection by RNA viruses. Taken together, these data demonstrate an essential role for PARP9 in type I IFN production in mouse BMDC and BMDM in response to intracellular poly I:C and RNA viruses infection.

**PARP9 protects mice against infection by RNA viruses in vivo.** To further elucidate the importance of PARP9 in vivo in mediating host defense against infection by RNA viruses, we first intraperitoneally infected both WT and PARP9 KO mice with the

RNA virus VSV and monitored survival over time. We found that the mortality of these PARP9 KO mice was significantly higher than that of their WT littermates (Fig. 3a). In addition, the qPCR analysis showed that the expression of Ifnb1 in liver, lungs, spleen, and brain of PARP9 KO mice was significantly lower than that in those organs from WT littermates, after VSV infection (Fig. 3b). In contrast, VSV replication and titers in liver, lung, spleen, and brain were significantly higher in PARP9 KO mice than in their WT counterparts (Fig. 3c and d). In addition, PARP9 KO mice produced much less type I IFN in sera than did their PARP9-sufficient littermates after VSV infection (Fig. 3e), while the inflammatory cytokines IL-6, TNF-α, and IL-10 in sera (Supplementary Fig. S7a) and the cell composition of CD4$^+$ T, CD8$^+$ T and B cells in spleen (Supplementary Fig. S7b and c) were comparable for WT and PARP9 KO mice at day 1 after VSV infection. Furthermore, lung histopathology revealed there was more infiltration of inflammatory cells into lungs of PARP9 KO mice than in those of their PARP9-sufficient counterparts following VSV infection (Fig. 3f). Importantly, both VSV infected WT and PARP9 KO mice survived much worse after treatment

with anti-IFNAR antibody compared to those treatment with isotype control antibody (Fig. 3g). In addition, PARP9 in CD11c + DCs isolated from WT mice with VSV infection and isotype treatment was dramatically induced by IFN with VSV infection, while PARP9 in those DCs isolated from WT mice with VSV infection and anti-IFNAR antibody treatment was significantly reduced due to IFNAR blockage (Fig. 3h), further suggesting PARP9 plays a role mainly in the positive feedback loop of IFN induction. By contrast, PARP9 played no role on mice survival in response to DNA virus HSV-1 infection in vivo (Fig. 3i).

Next, we also infected both WT and PARP9 KO mice intraperitoneally with dsRNA virus reovirus and monitored survival over time. We found that PARP9 KO mice were more susceptible to reovirus infection than their WT littermates (Supplementary Fig. S8a). In addition, we harvested heart, intestine, spleen, liver, and brain at day 2 post-infection for determining viral titers. We detected significantly more reovirus loads in PARP9 KO mice than in their WT littermates (Supplementary Fig. S8b). As expected, compared with WT mice, PARP9 KO mice produced three- to fourfold less type I IFN following reovirus infection (Supplementary Fig. S8c and d). Importantly, liver histopathology revealed the liver from PARP9 KO mice showed more severe diffuse necrosis and loss of structural markings compared to WT littermates at day 6 after reovirus infection (Supplementary Fig. 8e). Similarly, heart histology showed that the heart from PARP9 KO mice showed more severe inflammatory lesions and infiltrated cells, consistent with severe myocarditis, compared to WT littermates at day 6 after reovirus infection (Supplementary Fig. S8f). Together, these data demonstrate that PARP9 was required for efficient production of type I IFN in vivo and for the resistance of these mice to RNA virus infection.

**PARP9 recognizes and binds viral dsRNA.** To further explore the molecular mechanisms by which PARP9 induces type I IFN production after RNA virus infection, we first investigated whether PARP9 directly bind viral genomic dsRNA from reovirus (ReoRNA) or dsRNA poly I:C. We prepared recombinant hemagglutinin (HA)-tagged PARP9 by transfecting HEK 293T cells with plasmid HA-PARP9 and then purifying them with anti-HA beads. The purified HA-tagged PARP9 was then incubated with biotinylated poly (dA:dT), biotinylated poly (I:C) or biotinylated ReoRNA. Viral dsRNA ReoRNA (strong) and poly (I:C) (weak), but not DNA poly (dA:dT), efficiently precipitated HA-tagged PARP9 (Fig. 4a), suggesting that PARP9 prefers to bind viral dsRNA. However, the known viral RNA binding protein PARP13[37] in PARP family could only bind very weakly to viral dsRNA ReoRNA but not for poly (I:C) (Fig. 4a). To further investigate if viral dsRNA ReoRNA specifically bound to PARP9, we prepared and purified HA-tagged PARP9, PARP3, and PARP14, and incubated them with biotinylated ReoRNA. The viral dsRNA ReoRNA specifically precipitated HA-tagged PARP9, but not PARP3 or PARP14 (Fig. 4b). In addition, free viral dsRNA ReoRNA and dsRNA mimic poly (I:C), but not single RNA poly(A), DNA poly(dA:dT), and CpG-B, could compete with bioinylated ReoRNA binding to HA-tagged PARP9 (Fig. 4c and Supplementary Fig. S9a). PARP9 contains three domains including Macro A, Macro B and adenosine diphosphate (ADP) (Fig. 4d). To further map the viral dsRNA-binding domain of PARP9, we prepared and purified HA-tagged PARP9 deletion mutants (Fig. 4d). The biotinylated viral dsRNA ReoRNA co-precipitated with HA-tagged PARP9 deletion mutants C, N, and ΔADP, but not ΔMA, ΔMB, and MB, suggesting that the Macro domain of PARP9 was required for viral dsRNA ReoRNA binding (Fig. 4e). These data indicate that

PARP9 has viral dsRNA binding activity and the Macro domain is responsible for this function.

To further identify the physiological RNA species recognized and bound by PARP9 during infection with RNA virus reovirus, we purified the RNAs that immunoprecipitated together with anti-PARP9 in WT and PARP9 KO BMDC left infected (Mock) or infected with reovirus (Supplementary Fig. S9b). Since expression of PARP9 was very low in human HEK 293T cells, HA-PARP9 was then overexpressed for sensing RNA and triggering the activation of IFN-β promoter pathway in HEK 293T cells by luciferase reporter assay. The PARP9-bound RNA from reovirus-infected WT BMDC induced robust activation of an IFN-β-driven promoter in HEK 293T cells overexpressing HA-PARP9, but the PARP9 bound RNA from reovirus-infected PARP9 KO BMDC did not (Supplementary Fig. S9c). In contrast, the immunostimulatory activity of PARP9-bound RNAs extracted from uninfected (Mock) BMDC was minimal (Supplementary Fig. 9c), which indicated emergence of PARP9 ligands during infection with reovirus. The RNA gel analysis showed that there were specific RNAs from 1000 bp to 2000bp RNAs extracted from reovirus-infected WT BMDC, but not those from uninfected (Mock) WT BMDC (Supplementary Fig. S9d). Those specific RNAs was digested by low salt concentration Rnase A and Rnase III, but not Dnase I, Rnase H and high salt concentration Rnase A (Supplementary Fig. S9d), suggesting that the specific RNAs bound by PARP9 from reovirus-infected BMDC belonged to dsRNA. Next, we analyzed PARP9-bound RNAs extracted from WT and PARP9 KO BMDC left infected (Mock) or infected with reovirus by RNAseq and mapped the resulting sequences to both the reovirus genome and the mouse genome. This analysis revealed that the dsRNA mapping rate of PARP9-bound RNAs from reovirus-infected WT BMDC was up to 0.3% (Supplementary Fig. S9e). Importantly, the physiological dsRNA species recognized by PARP9 during reovirus infection were 8 to 1187 bp of S4 gene (Reo1187), 11 to 1198 bp of S3 gene (Reo1198), 16 bp to 1320 bp of S2 gene (Reo1320), and 9 to 1410 bp of S1 gene (Reo1410) (Supplementary Fig. S9f). We then synthesized in vitro those four dsRNA recognized by PARP9 and investigated their immunostimulatory activity by luciferase reporter assay. The Reo1198 induced significantly robust activation of an IFN-β-driven promoter in HEK 293T cells overexpressing HA-PARP9, but the Reo1187, Reo1320, and Reo1410 did not (Supplementary Fig. S9g). To more definitively determine whether PARP9 directly binds viral dsRNA Reo1198, the cell-free recombinant PARP9 was incubated with biotinylated Reo1198. Biotinylated Reo1198 robustly bound recombinant PARP9 protein (Fig. 4f). The electrophoretic mobility shift assay (EMSA) showed that recombinant PARP9 bound to biotinylated Reo1198 in a dose-dependent manner, free dsRNA Reo1198 could compete with biotinylated Reo1198 binding to recombinant PARP9 (Fig. 4g). Furthermore, full-length PARP9 and its deletion mutants C, N, and ΔADP, but not ΔMA, ΔMB and MB mutants, bound to biotinylated Reo1198 (Fig. 4g). We next determined if overexpression of full-length PARP9 or its mutant ΔMA could rescue the phenotype of PARP9 KO BMDC. Both full-length and mutant ΔMA of PARP9 were overexpressed in PARP9 KO BMDC (Supplementary Fig. S10a) by transducing with a lentiviral vector expressing HA-tagged full PARP9 and mutant ΔMA. As expected, overexpression of full-length PARP9, but not mutant PARP9 ΔMA, in PARP9 KO BMDC produced significantly more IFN-α (Supplementary Fig. S10b and c) than those cells expressing control vector in response to dsRNA LPIC and SPIC (Supplementary Fig. S10b) or RNA viruses reovirus and VSV (Supplementary Fig. S10c), suggesting that the macro domain of PARP9 is essential for binding viral dsRNA and inducing type I IFN after RNA virus infection. Taken together, these data

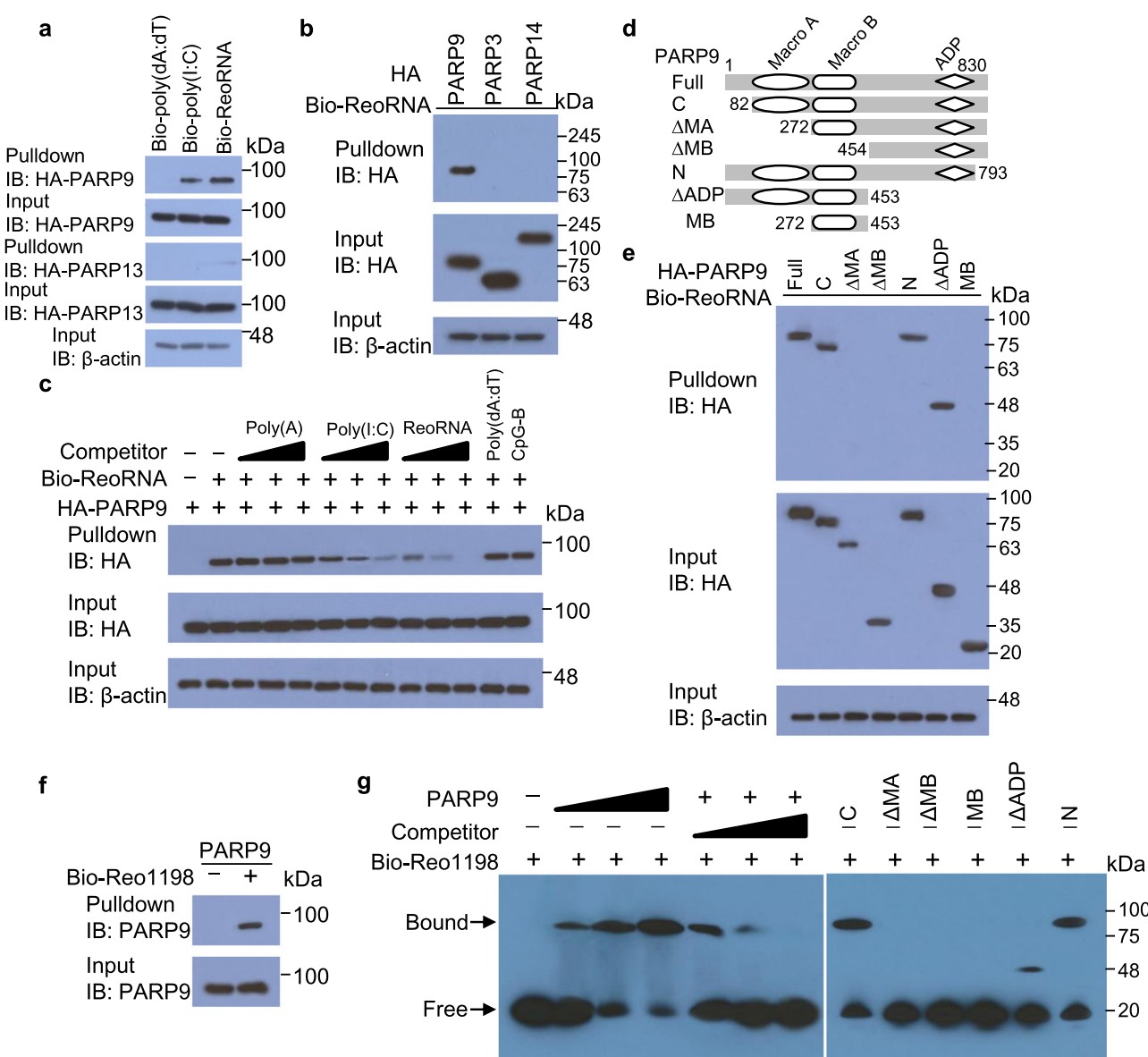

**Fig. 4 PARP9 physically interacts with viral dsRNA. a** Streptavidin pull-down assays of the binding of biotinylated poly(dA:dT) (Bio-poly(dA:dT)), biotinylated long poly(I:C) (Bio-poly(I:C)), or biotinylated reovirus dsRNA (Bio-ReoRNA) to purified HA-tagged PARP9 or PARP13 in HEK 293T cells. **b** Streptavidin pull-down assays of the binding of Bio-ReoRNA to indicated purified HA-tagged proteins in HEK 293T cells. **c** Streptavidin pull-down assays of the binding of Bio-ReoRNA to purified HA-tagged PARP9 without (−) or with competitors poly (A), poly (I:C) or ReoRNA (increasing concentrations of 0.5, 5, and 50 μg/ml), poly(dA:dT), CpG B. **d** Schematic diagram showing full-length PARP9 (Full) and serial truncations of PARP9 with deletion (Δ) of various domain (left margin); numbers at ends indicate amino acid positions (top). ADP, adenosine diphosphate. **e** Streptavidin pull-down assays of the binding of Bio-ReoRNA to indicated purified HA-tagged full-length and various truncatants of PARP9 in HEK 293T cells. **f** Streptavidin pull-down assays of the binding of biotinylated reovirus dsRNA (Bio-Reo1198) to cell-free, recombinant PARP9. **g** Electrophoretic mobility shift assay (EMSA) of Bio-Reo1198 with cell-free, recombinant full length PARP9 (5 μg or increasing concentrations of 0.5, 1, and 5 μg) or its mutants (5 μg) without or with competitor unlabeled viral dsRNA Reo1198 at 5, 10, and 20 times molar excess. Data are representative of three independent experiments.

demonstrate that physiological RNA ligands recognized and bound preferably by PARP9 are viral dsRNA from 1100 bp to 1400 bp during RNA virus infection.

**MAVS is dispensable in PARP9 induced production of type I IFN.** RLRs are major RNA sensors during RNA virus infections. Most RNA sensors trigger the type I IFN production through the downstream adapter protein MAVS[18–20]. To assess whether MAVS was also involved in PARP9-mediated production of type I IFN after sensing viral RNA, we examined the promoter activities of IFN-β in MAVS KO HEK 293T cells[38] overexpressing HA-PARP9 (luciferase reporter assay) in response to stimulation

with LPIC, viral dsRNA Reo1198, reovirus or influenza virus infection. Overexpression of PARP9 induced robust activation of IFN-β-driven promoter in MAVS KO HEK 293T cells in response to viral dsRNA Reo1198, reovirus or influenza virus infection (Fig. 5a), suggesting that PARP9 mediated IFN-β promoter activities in HEK 293T cells are MAVS-independent. To further confirm if PARP9 triggers MAVS-independent production of type I IFN in innate immune cells, we generated PARP9 and MAVS double knockout (DKO) mice by cross-breeding PARP9 KO mice with MAVS KO mice and the deletion of both PARP9 and MAVS was confirmed by genotyping PCR (Supplementary Fig. S11a) and immunoblot analysis in BMDC (Fig. 5b).

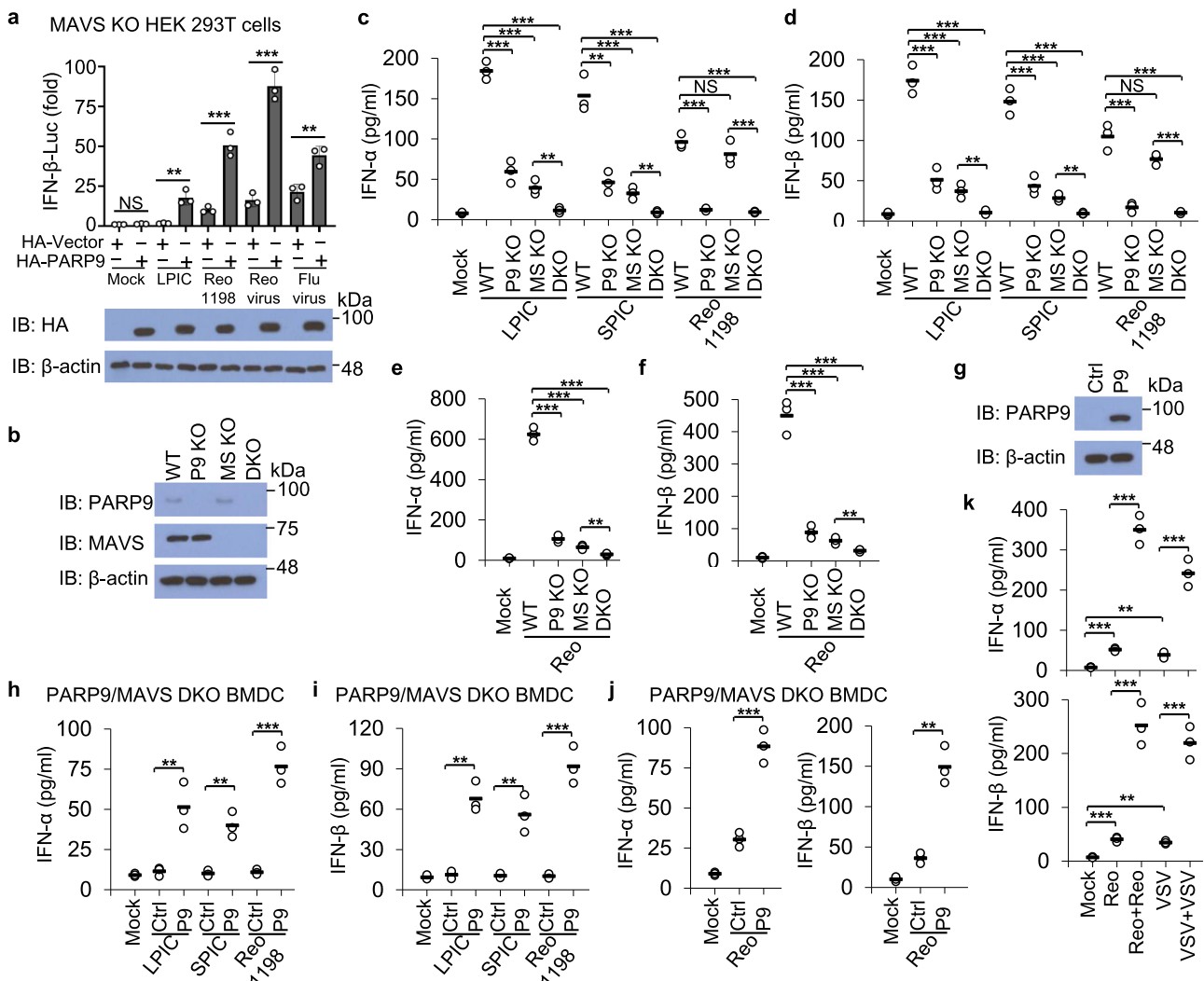

**Fig. 5 PARP9 initiates MAVS-independent production of type I IFN. a** IFN-β-Luciferase reporter assays and immunoblot (IB) analysis of HA-tagged PARP9 expression in MAVS knockout (KO) HEK 293 T cells transfected with IFN-β-Luc reporter plasmid, plus HA vector or HA-PARP9 plasmid for 12 h, followed by 10 h of stimulation without (Mock) or with long poly I:C (LPIC, 0.5 μg/ml) and viral dsRNA Reo1198 (Reo1198, 0.5 μg/ml) delivered by Lipofectamine 3000 or infection with reovirus or influenza (Flu) virus at MOI of 5 (n = 3 per group). Results are presented relative to those from unstimulated cells, set as 1. Error bars indicate standard error of the mean. **b** Immunoblot analysis of PARP9 or MAVS in mouse BDMC from wild-type (WT), PARP9 knockout (P9 KO), MAVS knockout (MS KO), and PARP9/MAVS double knockout (DKO) mice. ELISA of IFN-α (**c**) and IFN-β (**d**) production in mouse BDMC from wild-type (WT), PARP9 knockout (P9 KO), MAVS knockout (MS KO), and PARP9/MAVS double knockout (DKO) mice after 10 h of stimulation with long poly I:C (LPIC, 0.5 μg/ml), short poly I:C (SPIC, 0.5 μg/ml) or viral dsRNA Reo1198 (Reo1198, 1 μg/ml) delivered by Lipofectamine 3000 (n = 3 per group). ELISA of IFN-α (**e**) and IFN-β (**f**) production in mouse BDMC from WT, P9 KO, MS KO, and DKO mice after 12 h of infection with reovirus at MOI of 5 (n = 3 per group). **g** Immunoblot analysis of PARP9 expression in PARP9/MAVS double knockout (DKO) BMDC reconstituted with control vector (Ctrl) or wild-type PARP9 (P9). ELISA of IFN-α (**h**) and IFN-β (**i**) production by DKO BMDC reconstituted with control vector (Ctrl) or wild-type PARP9 (P9) after 10 h of stimulation with long poly I:C (LPIC, 0.5 μg/ml), short poly I:C (SPIC, 0.5 μg/ml), or viral dsRNA Reo1198 (Reo1198, 1 μg/ml) delivered by Lipofectamine 3000 (n = 3 per group). **j** ELISA of IFN-α and IFN-β production by DKO BMDC reconstituted with control vector (Ctrl) or wild-type PARP9 (P9) after 12 h of infection with reovirus at MOI of 5 (n = 3 per group). **k** ELISA of IFN-α and IFN-β production by MAVS KO BMDC after 6 h of first infection with reovirus, VSV at MOI of 5 or rechallenge with reovirus and VSV for another 6 h after first infection (n = 3 per group). Each circle represents an individual independent experiment and small solid black lines indicate the average of triplicates for results in **c–f** and **h–k**. NS, not significant (p > 0.05), **p < 0.01, ***p < 0.001, and p value was calculated by unpaired two-tailed Student's t test. Mock, BMDC without stimulation or infection. Data are representative of three independent experiments. Exact p values (**a**, p = 0.057, p = 0.0037, p = 0.00085, p = 0.00029, p = 0.0062; **c**, p = 0.0003, p = 0.00006, p = 0.00001, p = 0.0071, p = 0.0021, p = 0.00099, p = 0.0004, p = 0.0058, p = 0.00008, p = 0.057, p = 0.00007, p = 0.0004; **d**, p = 0.0007, p = 0.0003, p = 0.0001, p = 0.0074, p = 0.0008, p = 0.0003, p = 0.0001, p = 0.0017, p = 0.0009, p = 0.052, p = 0.0006, p = 0.0001; **e** p = 0.00002, p = 0.00001, p < 0.00001, p = 0.0071; **f** p = 0.0004, p = 0.0002, p = 0.0002, p = 0.0078; **h**, p = 0.0095, p = 0.0031, p = 0.00063; **i** p = 0.0011, p = 0.005, p = 0.00054; **j**, left, p = 0.00087, right, p = 0.0013; **k** upper, p = 0.00014, p = 0.0023, p = 0.0005, p = 0.0005, lower, p = 0.0008, p = 0.0015, p = 0.0005, p = 0.0009).

Compared to WT BMDC, PARP9 KO or MAVS KO or both PARP9 and MAVS DKO reduced dramatically production of IFN-α (Fig. 5c and e) and IFN-β (Fig. 5d and f) in PARP9 KO or MAVS KO BMDC in response to stimulation with intracellular poly (I:C) (Fig. 5c and d) or reovirus infection (Fig. 5e and f). Compared to PARP9 and MAVS DKO BMDC, MAVS KO produced significantly more IFN-α (Figs. 5c and 5e) and IFN-β (Fig. 5d and f) in response to stimulation with intracellular poly (I:C) (Fig. 5c and d) or reovirus infection (Fig. 5e and f). Importantly, for PARP9 specific ligand Reo1198 stimulation, PARP9 KO or both PARP9 and MAVS DKO reduced production of IFN-α (Fig. 5c) and IFN-β (Fig. 5d), while MAVS KO did not affect type I IFN production compared to WT BMDC (Fig. 5c and d). These data indicated that PARP9 could induce MAVS-independent production of type I IFN in BMDC after sensing viral dsRNA Reo1198 or reovirus infection. Additionally, MAVS KO or both PARP9 and MAVS DKO abrogated IL-6 production in response to intracellular poly I:C or reovirus infection, while PARP9 KO did not affect IL-6 production compared to WT BMDC (Supplementary Fig. S11b and c).

Next, we investigated if overexpression of PARP9 could induce type I IFN production in PARP9 and MAVS DKO BMDC after sensing dsRNA or reovirus infection. PARP9 was overexpressed in PARP9 and MAVS DKO BMDC (Fig. 5g) by transducing with a lentiviral vector expressing wild-type PARP9. As expected, overexpression of PARP9 in PARP9 and MAVS DKO BMDC produced significantly more IFN-α (Fig. 5h and j) and IFN-β (Fig. 5i and j) than those cells expressing control vector in response to intracellular poly I:C or viral dsRNA Reo1198 (Fig. 5h and i) or reovirus infection (Fig. 5j). However, overexpression of PARP9 in PARP9 and MAVS DKO BMDC did not affect the IL-6 production in response to intracellular poly I:C or viral dsRNA Reo1198 (Supplementary Fig. S11d) or reovirus infection (Supplementary Fig. S11e). Since PARP9 expression was low in MAVS KO BMDC and there were few type I IFN induction in MAVS KO BMDC after RNA virus infection, we further investigated if induced PARP9 could induce MAVS-independent more type I IFN production in MAVS KO BMDC after twice challenge with RNA viruses. Indeed, there were a little induction of IFN-α and IFN-β production in MAVS KO BMDC after first round of infection with reovirus or VSV (Fig. 5k). However, more significant production of IFN-α and IFN-β production was detected in MAVS KO BMDC after two rounds of challenge with reovirus or VSV (Fig. 5k), suggesting that type I IFN production induced by PARP9 after RNA virus infection was entirely independent of MAVS. We next determined the susceptibility of WT, PARP9 KO, MAVS KO, and PARP9/MAVS DKO mice in response to VSV infection. Compared to WT littermates, mortality of PARP9 KO, MAVS KO, or DKO mice was significantly higher after VSV infection (Supplementary Fig. S12). Notably, MAVS KO mice survived much better than DKO mice after VSV infection, suggesting that PARP9 plays an important role in host defense against VSV infection by acting independent of the MAVS pathway.

Given that PARP9 was very lowly expressed and induced by IFN-α in human monocyte-derived dendritic cells (MDDC), we firstly treated human MDDC with or without MAVS or PARP9 knockdown using IFN-α for 2 h and then stimulated those cells with intracellular LPIC or VSV infection. MAVS was indeed knocked down by MAVS-targeting shRNA, while IFN-α treatment-induced dramatically the expression of PARP9 compared to the cells without treatment (Supplementary Fig. S13a). Knockdown of MAVS or both MAVS and PARP9 abrogated almost the production of IFN-β compared to the control human MDDC in response to intracellular LPIC or VSV infection (Supplementary Fig. S13b). The MAVS knockdown human MDDC with IFN-α

treatment produced more IFN-β than those cells without IFN-α treatment (Supplementary Fig. S13b), suggesting PAPR9 initiates and amplifies MAVS-independent induction of type I IFN in human MDDC. Taken together, these data demonstrate PARP9 indeed induces MAVS-independent production of type I IFN after sensing RNA virus infection in both human and mouse DCs.

**PARP9 binds to and activates the downstream PI3K p85.** To further explore the MAVS-independent signal transduction pathway induced by PARP9, we used an antibody specific to PARP9 to immunoprecipitate PARP9-interacting proteins in cell lysates from human THP1 macrophages, followed by protein sequencing by liquid chromatography-mass spectrometry. Among a group of PARP9-interacting proteins, we found mammalian target of rapamycin (mTOR) was among a group of PARP9-interacting proteins (Supplementary Table S1). We next investigated if PARP9 could interact with mTOR and PI3K in BMDC at the endogenous protein level. The anti-PARP9 antibody, but not control IgG, precipitated mTOR, but not PI3K regulatory subunit p85, in uninfected (Mock) BMDC (Fig. 6a). However, the interaction between PARP9 and mTOR was disrupted after VSV infection (Fig. 6a), but the strong interaction between PARP9 and PI3K p85 was detected after VSV infection, indicating real interaction between PARP9 and PI3K p85 in BMDC after VSV infection (Fig. 6a). To further map the binding sites between PARP9 and p85, we analyzed interactions among Myc-tagged recombinant p85 and HA-tagged recombinant full-length PARP9, as well as truncation mutants of PARP9 (Fig. 4d). Both full-length PARP9 and the ADP domain of PARP9 bound to p85 (Fig. 6b). Additionally, the mapping results for Myc-tagged recombinant PARP9 and HA-tagged full-length p85 and their truncation mutants (Fig. 6c) showed that the SH2 (Src Homology 2) domain of p85 bound to PARP9 (Fig. 6d). Furthermore, immunofluorescence of PARP9 and p85 showed that endogenous PARP9 colocalized with endogenous p85 in the cytosol in primary peritoneal macrophages after VSV infection at 2 h (Fig. 6e and f). By contrast, there was no colocalization between p85 and PARP9 without viral infection (Fig. 6e and f). Similarly, no colocalization of PARP9 and TBK1 was detected after VSV infection (Fig. 6e and f). Importantly, strong interaction between PARP9 and p85 was also detected in human MDDC after reovirus dsRNA stimulation, indicating binding of viral RNA could lead to activation of PARP9 and interaction with PI3K p85 in both human and mouse DCs (Fig. 6g).

Furthermore, we investigated activation of PI3K after VSV or reovirus infection by detecting phosphorylation of p85. PARP9 KO BMDC exhibited more reduced p85 phosphorylation compared to WT BMDC after VSV or reovirus infection (Fig. 6h). We next conducted the in vitro p85 kinase assay by measuring phosphatidylinositol 3,4,5-trisphosphate (PIP3), which is generated when the substrate phosphatidylinositol 4,5-bisphosphate (PIP2) is converted by activated p85. Indeed, PARP9 KO reduced significantly the relative amount of PIP3 in PARP9 KO BMDC infected by VSV or reovirus (Fig. 6i), indicating a reduction of PI3K activity. Collectively, these data suggested that PARP9 interacts with p85 for triggering downstream activation of PI3K p85 pathway.

**PI3K p85 triggers downstream AKT3 activation to phosphorylate IRF3 and IRF7 for type I IFN production.** The AKT family members, including AKT1, AKT2, AKT3, are the downstream targets of PI3K activation[39]. To investigate which one of AKT family members is the real target of PI3K activation, we detected AKT family members in BMDC after VSV infection. Interestingly, expression of both AKT1 and AKT2 at mRNA level was

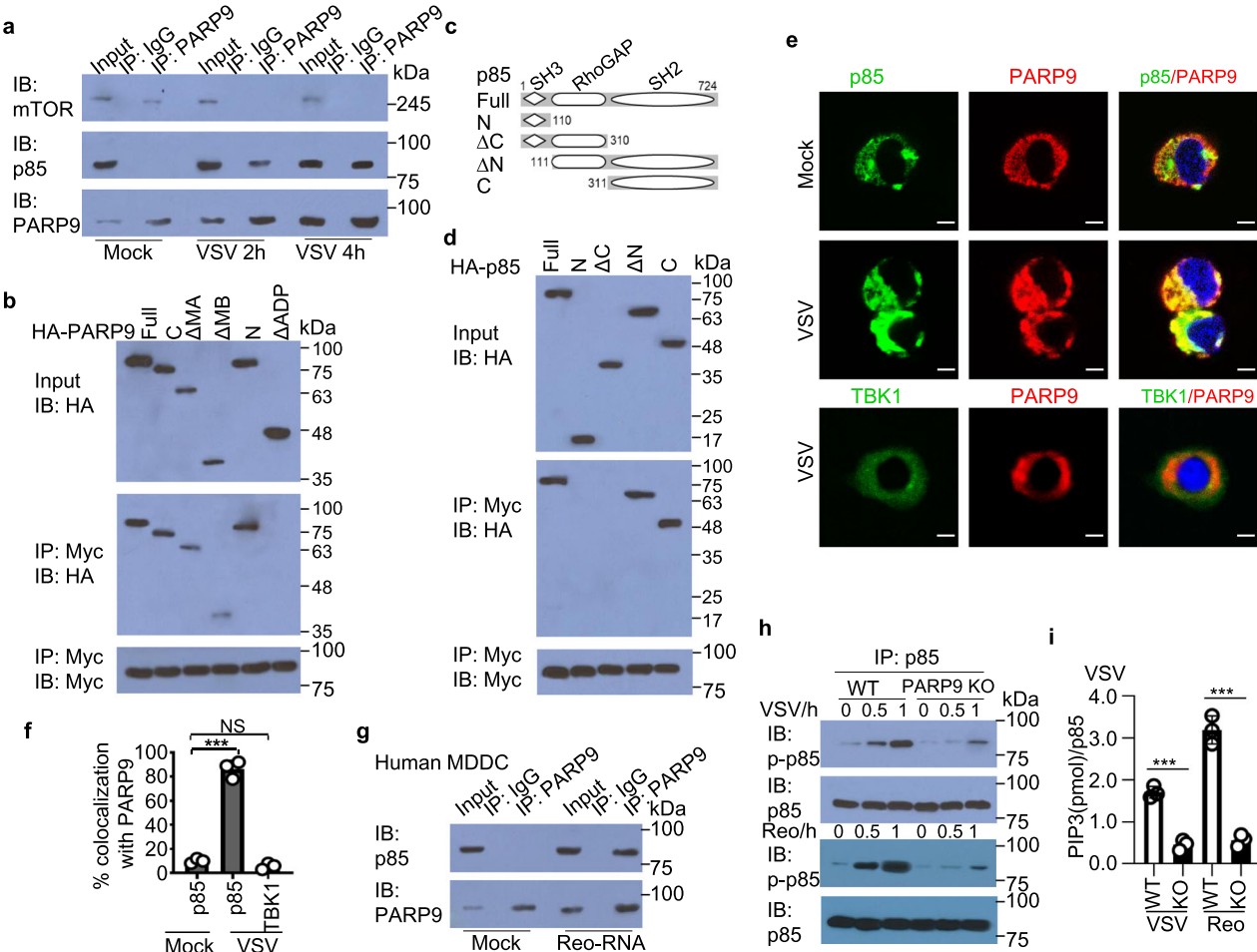

**Fig. 6 PARP9 recruits and activates downstream PI3K p85. a** Immunoblot analysis of endogenous proteins PARP9, mammalian target of rapamycin (mTOR) and phosphoinositide 3-kinase (PI3K) regulatory subunit p85 precipitated with anti-PARP9, or immunoglobulin G (IgG) from whole-cell lysates of wild-type BMDC left infected (Mock) or infected with VSV at MOI of 1 for 2 h or 4 h. **b** Immunoblot analysis of purified Myc-tagged p85 with anti-Myc antibody (bottom blot), and immunoblot analysis of purified HA-tagged full-length PARP9 and serial truncation of PARP9 with deletion (Δ) of various domains alone with anti-HA antibody (top blot) or after incubation with Myc-tagged p85 and immunoprecipitation with anti-Myc antibody (middle blot). **c** Schematic diagram showing full-length p85 (Full) and serial truncations of p85 with deletion (Δ) of various domain (left margin); numbers at ends indicate amino acid positions (top). SH3, the Src-homology 3 domain; RhoGAP, Rho GTPase activating protein domain; SH2, the Src-homology 2 domain. **d** Immunoblot analysis of purified Myc-tagged PARP9 with anti-Myc antibody (bottom blot), and immunoblot analysis of purified HA-tagged full-length p85 and serial truncations of p85 with deletion of various domains alone with anti-HA antibody (top blot) or after incubation with Myc-tagged PARP9 and immunoprecipitation with anti-Myc antibody (middle blot). **e** Confocal microscopy of primary peritoneal macrophages from WT mice left infected (Mock) or infected with VSV at MOI of 1 for 2 h. p85 or TBK1 was stained with mouse anti-PI3 kinase p85 alpha monoclonal antibody (Cat: ab86714, Abcam) or mouse anti-TBK1 monoclonal antibody (Cat: NB100-56705, Novus Biologicals), followed by Alexa Fluor 488 goat anti-mouse secondary antibody (green), while PARP9 was stained with rabbit anti-PARP9 polyclonal antibody (Cat: LS-B9440, LifeSpan BioSciences), followed by Alexa Fluor 594 goat anti-rabbit secondary antibody (red). DAPI (4′,6-diamidino-2-phenylindole) served as the nuclei marker (blue). Scale bars represent 10 μm. **f** Quantification of colocalization between PARP9 and p85 or TBK1 using ImageJ software (n = 3 per group). **g** Immunoblot analysis of endogenous proteins PARP9 and PI3K regulatory subunit p85 precipitated with anti-PARP9, or immunoglobulin G (IgG) from whole-cell lysates of human MDDC left transfected (Mock) or transfected with reovirus dsRNA (Reo-RNA) for 6 h. **h** Immunoblot analysis of endogenous proteins p85 and phosphorylated p85 (p-p85) precipitated with anti-p85 from whole-cell lysates of wild-type (WT) or PARP9 KO BMDC left infected (0 h) or infected with VSV or reovirus (Reo) at MOI of 1 for 0.5 or 1 h. **i** The relative amounts of PIP3 in the plasma membrane were normalized to the amount of immunoprecipitated p85 in WT and PARP9 KO BMDC after VSV infection (n = 3 per group). Error bars indicate standard error of the mean for results in **f**, **i**. NS, not significant (p > 0.05), ***p < 0.001, and p value was calculated by unpaired two-tailed Student's t test. Data are representative of three independent experiments. Exact p values (**f**, p = 0.00007, p = 0.08; **i**, p = 0.0003, p = 0.00024).

reduced after VSV or reovirus infection (Fig. 7a). However, expression of AKT3 was markedly induced in BMDC upon VSV or reovirus infection (Fig. 7a), indicating AKT3 maybe the downstream target of PI3K p85 activation. We next knocked down AKT1, AKT2, and AKT3 and examined the expression of IFN-β in BMDC followed by VSV or reovirus infection. We confirmed that the AKT1, AKT2, or AKT3 were efficiently knocked down in VSV-infected BMDC (Fig. 7b). Knockdown of

AKT3, but not AKT1 or AKT2, substantially reduced the enhanced expression of IFN-β at mRNA level in BMDC infected by VSV or reovirus (Fig. 7c), suggesting that AKT3 is the possible downstream target of PI3K activation after VSV or reovirus infection. To further confirm if AKT3 is real target of PI3K activation, we detected the interaction of PI3K p85 with those three AKT family members in VSV-infected BMDC (Fig. 7d). Indeed, PI3K p85 could interact with AKT3, but not AKT1 and

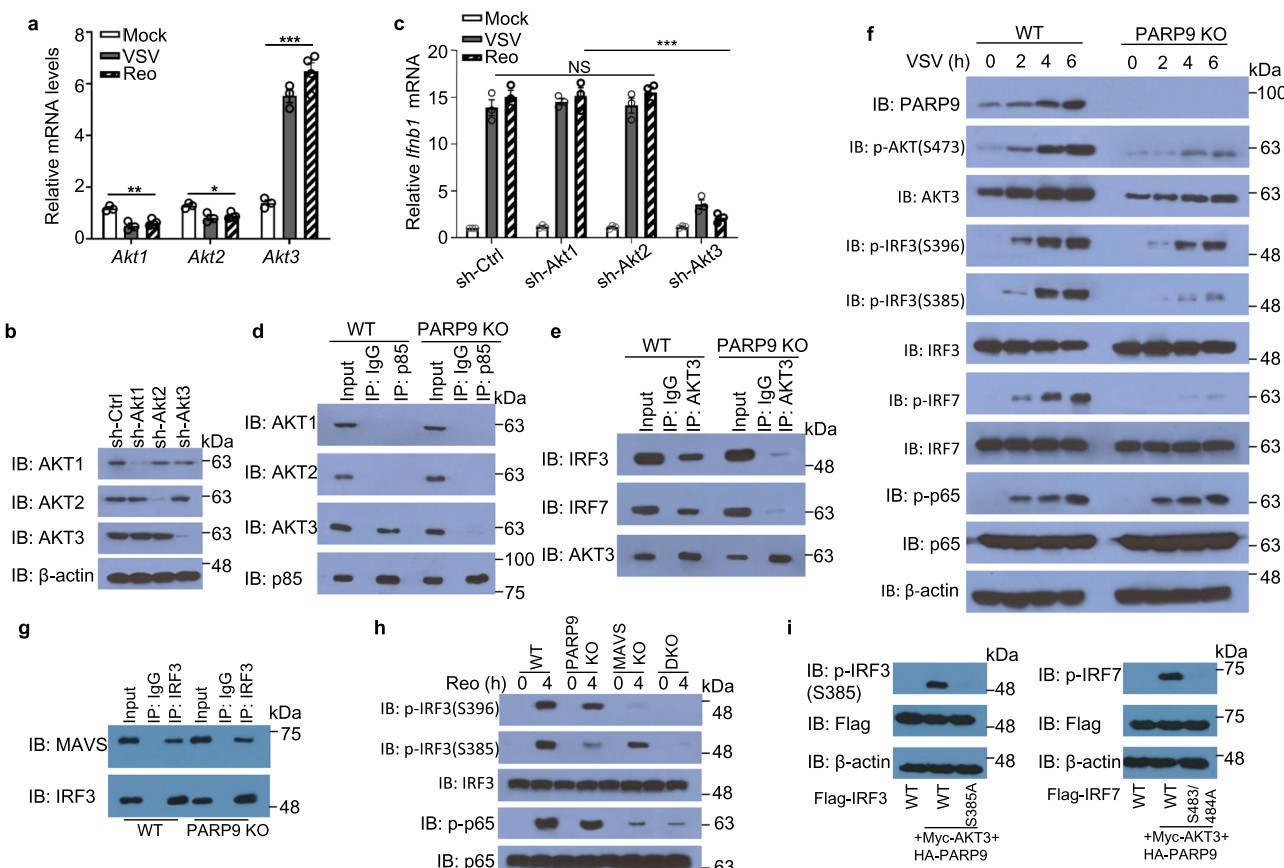

**Fig. 7 PI3K p85 activates downstream AKT3 to directly phosphorylate IRF3 and IRF7 to induce type I IFN production. a** Relative mRNA levels of Akt1, Akt2, and Akt3 in wild-type BMDC left infected (Mock) or infected with VSV or reovirus (Reo) at MOI of 1 for 2 h (n = 3 per group). **b** Immunoblot analysis of AKT1, AKT2, or AKT3 in wild-type BMDC treated with shRNA to knockdown expression of AKT1, AKT2 or AKT3, followed by VSV infection at MOI of 1 for 2 h. A scrambled shRNA served as a control (sh-Ctrl). The β-actin served as the loading control. **c** Relative *Ifnb1* mRNA levels in wild-type BMDC treated with the indicated shRNA, followed by a 2 h infection without (Mock) or with VSV or reovirus (Reo) at MOI of 1 (n = 3 per group). **d** Immunoblot analysis of endogenous proteins AKT1, AKT2, AKT3, and p85 precipitated with anti-p85, or immunoglobulin G (IgG) from whole-cell lysates of wild-type (WT) or PARP9 KO BMDC infected with VSV at MOI of 1 for 2 h. **e** Immunoblot analysis of endogenous proteins IRF3, IRF7 and AKT3 precipitated with anti-AKT3 from whole-cell lysates of wild-type (WT) or PARP9 KO BMDC infected with VSV at MOI of 1 for 2 h. **f** Immunoblot analysis of the PARP9, phosphorylated p65 (p-p65), p65, phosphorylated IRF3 at S396 (p-IRF3 S396), phosphorylated IRF3 at S385 (p-IRF3 S385), total IRF3, phosphorylated IRF7 at S477 (p-IRF7 S477), total IRF7, phosphorylated AKT at S473 (p-AKT S473), total AKT3, and β-actin in wild-type (WT) and PARP9 KO BMDC before (0) or various times (above lanes) after VSV infection at MOI of 1. **g** Immunoblot analysis of endogenous proteins MAVS and IRF3 precipitated with anti-IRF3, or immunoglobulin G (IgG) from whole-cell lysates of wild-type (WT) or PARP9 KO BMDC infected with reovirus at MOI of 1 for 2 h. **h** Immunoblot analysis of phosphorylated p65 (p-p65), p65, phosphorylated IRF3 at S396 (p-IRF3 S396), phosphorylated IRF3 at S385 (p-IRF3 S385), total IRF3 in wild-type (WT), PARP9 KO, MAVS KO, and PARP9/MAVS double knockout (DKO) BMDC before (0) or after reovirus (Reo) infection for 6 h at MOI of 1. **i** Immunoblot analysis of phosphorylated IRF3 (p-IRF3 S385), phosphorylated IRF7 (p-IRF7), Flag-IRF3 and Flag-IRF7 in HEK293T cells overexpressing WT IRF3, IRF3 mutant (S385A), WT IRF7, IRF7 mutant (S483/484 A corresponding to mouse IRF7 S437/438 A) together with Myc-AKT3 and HA-PARP9. Error bars indicate standard error of the mean for results in **a**, **c** . NS, not significant (p > 0.05), *p < 0.05, **p < 0.01, ***p < 0.001, and p value was calculated by unpaired two-tailed Student's t test. Mock, BMDC without infection. Data are representative of three independent experiments. Exact p values (**a**, p = 0.004, p = 0.02, p = 0.0001; **c**, p = 0.52, p = 0.0007).

AKT2, in VSV-infected WT BMDC (Fig. 7d). However, the interaction of p85 and AKT3 was disrupted in VSV-infected PARP9 KO BMDC (Fig. 7d). These data suggested that AKT3 is a real downstream target of PI3K p85 activation after virus infection. As IRF3 and IRF7 are important transcription factor to control type I IFN production, we next investigated if AKT3 could interact with IRF3 or IRF7. As expected, AKT3 could interact with IRF3 and IRF7 in VSV-infected WT BMDC (Fig. 7e). However, the interactions of AKT3 with IRF3 and IRF7 were dramatically reduced in VSV-infected PARP9 KO BMDC (Fig. 7e). Finally, we determined the downstream activation of signaling pathway triggered by PI3K p85 activation in both WT and PARP9 KO BMDC upon RNA virus VSV infection. AKT3 expression is very low in resting BMDC, but it is dramatically

induced in VSV-infected BMDC. For detecting AKT3 phosphorylation, we immunoprecipitated AKT3 from WT and PARP9 KO BMDC followed by immunoblot analysis using anti-AKT3 and anti-phosphorylated AKT antibody. As expected, PARP9 was significantly induced in WT BMDC upon VSV infection, but was absent in PARP9 KO BMDC (Fig. 7f). AKT3 was marked induced and phosphorylation of AKT was strongly activated in VSV-infected WT BMDC, however, AKT3 induction and AKT phosphorylation were significantly reduced in VSV-infected PARP9 KO BMDC (Fig. 7f). IRF3 Ser396 phosphorylation and IRF3 Ser385 phosphorylation were strongly activated in VSV-infected WT BMDC (Fig. 7f). By contrast, reduction of IRF3 Ser385 phosphorylation was significantly more than that of IRF3 Ser396 phosphorylation in VSV-infected PARP9 KO BMDC (Fig. 7f),

suggesting that downstream activations of PI3K p85 and AKT3 by PARP9 sensing RNA virus infection preferred to target and phosphorylate IRF3 at Ser385. Furthermore, IRF7 Ser437/438 phosphorylation was strongly activated in VSV-infected WT BMDC, however, the activation of IRF7 Ser437/438 phosphorylation reduced significantly in VSV-infected PARP9 KO BMDC (Fig. 7f). In contrast, the phosphorylation of NF-κB subunit p65 was comparable between VSV-infected WT and PARP9 KO BMDC (Fig. 7f). We next determined the interaction between IRF3 and MAVS in WT and PARP9KO BMDC infected by reovirus. As shown in Fig. 7g, this interaction is independent on PARP9 (Fig. 7g). Furthermore, we examined the phosphorylation of IRF3 and NF-κB subunit p65 in WT, PARP9 KO, MAVS KO, and PARP9/MAVS DKO BMDC after reovirus infection. Similarly, PARP9 preferred to target and phosphorylate IRF3 at Ser385 but not affect the phosphorylation of p65 in BMDC after reovirus infection (Fig. 7h), further confirming the above results from VSV infection. At last, we examined if loss of IRF3 phosphorylation site S385 or IRF7 phosphorylation sites S437/438 results in loss of PARP9/AKT3 signaling to IRF3 or IRF7. We then generated point mutations S385A IRF3, S483/484A IRF7 (corresponding to S437/438 of mouse IRF7), which were individually transfected into HEK293T cells together with HA-PARP9 and Myc-AKT3. PARP9 and AKT3 induced easily wild-type IRF3 phosphorylation at S385 or IRF7 phosphorylation at S483/484 (Fig. 7i), while either IRF3 point mutation S385A or IRF7 point mutation S483/484A blocked PARP9/AKT3 induced IRF3 phosphorylation or IRF7 phosphorylation (Fig. 7i). Collectively, these data showed that PI3K p85 activation by PARP9 targets AKT3 activation to directly phosphorylate IRF3 at Ser385 and IRF7 at Ser437/438 for type I IFN production after RNA virus infection.

## Discussion

PARP is a family of proteins with diverse roles in chromatin regulation, transcription, RNA biology, and DNA repair. Recent evidence extends their functions to include stress response, metabolism, viral infections, and cancer[21]. The multifunctional activities of PARP family proteins prompted us to investigate their roles in host innate immunity. In this study, we discovered that PARP9 serves as a non-canonical sensor for RNA virus infections, activating the innate immune cells via the MAVS-independent pathway. Also, PARP9 is highly induced by type I IFN, which in turn promotes and amplifies type I IFN response to RNA viruses, thus acting in a feed-forward manner. This is further supported in our PARP9 KO mice, where deletion of PARP9 in vivo rendered the PARP9 KO mice highly susceptible to infections with RNA viruses (e.g., VSV and reovirus) due to impaired type I IFN production in vivo. Mechanistically, we found that PARP9 binds to viral dsRNA, specifically in the region of viral dsRNA from 1100 to 1400 bp (Reo1187, Reo1198, Reo1320, and Reo1410). Importantly, almost all of the RNA sensors identified so far use the adapter protein MAVS to trigger type I IFN production[18–20], and our findings that PARP9-induced type I IFN production is independent of MAVS are significant. In fact, we are the first to generate PARP9 and MAVS double knockout (DKO) mice. Clearly, overexpression of PARP9 could induce type I IFN production in either MAVS KO HEK 293T cells or PARP9 and MAVS DKO BMDC after viral RNA stimulation or RNA virus infection, indicating that PARP9 acts as a non-canonical and MAVS-independent RNA sensor to trigger type I IFN production. Notably, PARP9 could be dramatically induced even with very low amount of type I IFN and induce robust amplification of type I IFN production after sensing viral dsRNA and RNA virus infection, suggesting that PARP9 mediates robust amplification of type I IFN production during RNA virus

infection. To further explore the downstream adapter of PARP9 after sensing RNA virus infection, we examined PARP9 interacting proteins by IP-MS (Supplementary Table S1) and found mTOR was among the PARP9 interacting compounds, which give us a clue to think about the PI3K/AKT/mTOR metabolic pathway. Indeed, PARP9 interacts with PI3K regulatory subunit p85 to trigger its phosphorylation and activation for type I IFN production, which are disrupted in PARP9 KO BMDC after RNA virus infection. Therefore, we propose a working model as seen below. In canonical model, after RNA virus infection, well-known RNA sensors RIG-I and MDA-5 recognize viral RNA and induce MAVS-dependent type I IFN production (Supplementary Fig. S14). RNA sensor TLR3 is identified by detecting extracellular viral nucleic acids during viral infection, while RNA sensor PKR does not signal directly to type I IFN production. In our non-canonical model, upon RNA virus infection, the non-canonical RNA sensor PARP9 senses viral RNA and induces MAVS-independent type I IFN response by recruiting and activating PI3K p85 and AKT3 pathway. The produced IFN-α could induce dramatically PARP9 expression and the induced PARP9 further enhances the type I IFN production and antiviral immune response (Supplementary Fig. S14). Overall, identification of PARP9 as a non-canonical and MAVS-independent RNA sensor might lead to strategies for intervention in RNA virus-induced diseases.

Macro domain family has the ancient, highly evolutionarily conserved macro domains that are widely distributed throughout all kingdoms of life ranging from viruses and bacteria to yeast and humans. There are at least 10 genes that encode 11 members of the macro domain family in human[40]. The wide distribution of the macro domain family suggests that it is involved in an important and ubiquitous cellular process and regulation of diverse biological functions, such as DNA repair, chromatin remodeling, and transcriptional regulation[40,41]. Until now, the role and precise regulatory mechanisms of macro domain family proteins in infectious disease remain largely uncharacterized. In this study, we have shown that PARP9 uses its macro domains for recognizing viral dsRNA from RNA viruses and employs PI3K p85 and AKT3 metabolic pathway to directly phosphorylate both IRF3 at Ser385 and IRF7 at Ser437/438 for producing type I IFN, which function to clear the RNA virus infections. Importantly, a large number of viruses and microbial parasites contain macro domain proteins, and some of these proteins are necessary for host cell infection and viral replication[21]. The non-structural protein (NSP3) macro domain has an essential role for sindbis virus replication and age-dependent susceptibility to encephalomyelitis[42]. Interestingly, the RNA viruses, such as SARS coronavirus, Hepatitis E virus, Chikungunya virus, and Venezuelan Equine Encephalitis virus, have the macro domain-containing NSP3 proteins. Interestingly, recent coronavirus-infected pneumonia epidemic in Wuhan, China is shown to be caused by SARS coronavirus 2[3]. So we speculate that these deadly SARS coronavirus 2 may employ its macro domain-containing NSP3 protein to target PARP9 and block its dsRNA recognition for shutting down type I IFN production and this strategy of innate immune evasion maybe the cause of SARS coronavirus infected disease.

The PI3K/AKT/mTOR pathway is also important in regulating adaptive immune cell activation[43,44]. Recently, it has increasingly been recognized that PI3K/AKT pathway has distinct roles in innate immune cells[44,45]. We are the first to show that PARP9 is activated by sensing viral dsRNA, and then recruits and activates PI3K/AKT3 pathway for producing type I IFN upon RNA virus infection. It has been reported that PI3K is critical for the nuclear translocation of IRF7 and type I IFN production by pDC in response to TLR agonists[46]. Previous studies show that PI3K is

critical in governing an effective IFN-α/β-mediated antiviral response and cells lacking p85α and p85β (p85α/β) show defective antiviral responses and reduce IFN-α/β-inducible ISG protein expression[47,48]. Our study shows that PI3K p85 activation triggered by PARP9 is essential for the phosphorylation of both IRF7 and IRF3 and type I IFN production in response to RNA virus infection. AKT family members include AKT1, AKT2, and AKT3[39]. AKT1 and AKT2 have been demonstrated to repress IFN-β production and support virus infection[49,50]. Another study suggests that AKT1 inhibits HSV-1 and VSV replication by phosphorylating EMSY, the BRCA2-interacting transcriptional repressor, to relieve EMSY-mediated ISG repression[51]. However, the role of AKT3 in innate immunity has not been studied thus far, and our study has revealed that PI3K p85 targets downstream AKT3 to directly phosphorylate both IRF3 at Ser385 and IRF7 at Ser437/438 for inducing type I IFN production. Most recently, AKT3 has been shown to directly bind and phosphorylated IRF3 at Ser385, together with TBK1-induced phosphorylation of IRF3 Ser386, to achieve IRF3 dimerization[52], which further confirms our results. We provide the first evidence that AKT3 is essential for type I IFN production during RNA virus infection.

Among the PARP family members, PARP13, also called ZAP (Zinc finger antiviral protein), PARP13 was reported to inhibit the production of retroviral RNA[27] and inhibit endogenous retrotransposition by long interspersed nuclear elements and Alu elements[53,54]. Moreover, it's reported that mice harboring the hyper-responsive form of STAT1 have been shown to enhance IFN signaling and control viral infection, and STAT1 employs E3 ubiquitin ligase deltex-3-like (DTX3L) through PARP9-DTX3L interaction to target host histone H2BJ and viral 3C protease[32]. Until now, no PARP9 KO mice is reported and whether PARP9 alone plays role in antiviral innate immunity is still elusive. In our study, we are the first to generate PARP9 KO mice and PARP9/MAVS DKO mice and demonstrate PARP9 alone, serving as a noncanonical and MAVS-independent RNA sensor, is essential for type I IFN production and plays an important role in antiviral host defense against infection with RNA viruses both in vitro and in vivo.

Importantly, our data show PARP9 signaling directs IRF3 and type I IFN antiviral response in a manner that does not highly engage the inflammatory signaling cascades directed by NF-κB. This aspect of the findings makes the study increasingly significant, and present a scenario in which one could consider therapeutic targeting of PARP9 signaling to treat virus infection and avoid enhancing an inflammatory cytokine storm.

In summary, we have identified PARP9 as a non-canonical RNA sensor for RNA viruses in initiating and amplifying protective immunity. Our results underscore the complexity of viral sensing mechanisms and the importance of PARP9 in triggering antiviral type I IFN production. Importantly, the 2019 coronavirus outbreak in Wuhan, China serves as a reminder that the new emerging RNA viruses remain a significant public health threat[3]. Thus, our findings on PARP9 may provide an opportunity for better interventions in boosting protective antiviral immunity through PARP9 induction with chemical compound. Our findings may also be beneficial in the design of better pharmacological agonists to improve the efficacy of RNA virus vaccines and prophylactics.

## Methods

**Mice.** *Parp9*[fl/fl] conditional knockout mice were generated at Taconic-Artemis (Cologne, Germany) in close consultation with our lab as follows: Mouse genomic fragments of the Parp9 locus were subcloned using RPCIB-731 BAC library via ET recombination and recloned into a basic targeting vector placing an F3-site flanked puromycin resistance cassette in intron 3 and a thymidine kinase cassette downstream of the 3' UTR. LoxP sites flanked exon 4. The targeting vector was sequenced to confirm correctness. The linearized DNA vector was electroporated into C57BL/6 N embryonic stem cells, puromycin selection (1 μg/ml) started on day 2 and counterselection with ganciclovir (2 μM) started on day 5 after electroporation. Embryonic stem cell clones were isolated on day 8 and analyzed by Southern blotting according to standard procedure. Blastocysts were isolated from the uterus of Balb/c females at day 3.5 post coitum and 10–15 targeted C57BL/6NTac embryonic stem cells were injected into each blastocyst. After recovery, six injected blastocysts were transferred to each uterine horn of 2.5 days post coitum, pseudopregnant females. Chimerism of offspring was measured by coat color contribution of embryonic stem cells to the Balb/c host (black/white). Highly chimeric mice were bred to strain C57BL/6 females transgenic for the Flp recombinase gene to remove the puromycin resistance cassette in mice carrying the conditional knockout allele (*Parp9*[fl/WT]), which were further self-crossed to generate *Parp9*[fl/fl] mice. Germline transmission was identified by the presence of black strain C57BL/6 offspring. The *Parp9*[fl/fl] mice were backcrossed to the C57BL/6 background strain over at least 6 generations before use in subsequent experiments.

The *Parp9*[fl/fl] mice were crossed with the *EIIa-Cre* transgenic mice (Stock No: 003314, The Jackson Laboratory) that express Cre recombinase in germline[55] to generate global knockout (*Parp9*[-/-]) mice for experiments. MAVS KO mice (Stock No: 008634, The Jackson Laboratory) were purchase from Jackson Laboratory. PARP9 KO mice were crossed with MAVS KO mice to generate PARP9 and MAVS double knockout (DKO, *Parp9*[-/-] *Mavs*[-/-]) mice, which were backcrossed to the C57BL/6 background strain over at least six generations before use in subsequent experiments. All animals were on the C57BL/6 genetic background and maintained in the specific pathogen-free facility under 12 h light/dark cycle at 22–24 °C with unrestricted access to food and water for the duration of the experiment at Houston Methodist Research Institute in Houston, Texas. Animal use and care were ethically approved by the Houston Methodist Animal Care Committee, in accordance with institutional animal care and use committee guidelines.

**Reagents.** The high molecular weight poly I:C (long poly I:C, Cat: tlrl-pic), low molecular weight poly I:C (short poly I:C, Cat: tlrl-picw), 5'triphosphate double-stranded RNA (5'pppRNA, Cat: tlrl-3prna), HSV-60 (Cat: tlrl-hsv60n), 2'3'-cGAMP (cGAMP, Cat: tlrl-nacga23), Poly(dA:dT) (Cat: tlrl-patn), Class B CpG (CpG-B, Cat: tlrl-1668) and Biotin-labeled poly(I:C) (Bio-poly(I:C), Cat: tlrl-picb) were from Invivogen. The poly (A) was from Sigma-aldrich. Lipofectamine 3000 (Cat: L3000015) was from Invitrogen. The following antibodies were used for immunoblot analysis: anti-PARP9 (IB:1:1000; IP:1:100; AB10618; Millipore), anti-PARP9 (IB:1:1000; LS-B9440; LifeSpan BioScience), mouse anti-PI3 kinase p85 alpha monoclonal antibody (IF:1:200; Cat: ab86714, Abcam), anti-MAVS (IB:1:1000; sc-166583; Santa Cruz), anti-IRF3 (IB:1:1000; sc-9082; FL-425; Santa Cruz), anti-IRF7 (IB:1:1000; AHP1180T, Bio-Rad), antibody to phosphorylated IRF3 at Ser396 (IB:1:1000; #4947; Cell Signaling Technology), antibody to phosphorylated IRF3 at Ser385 (IB:1:1000; D151514; Sangon Biotech), antibody to phosphorylated IRF7 at Ser437/438 (IB:1:1000; #24129S; Cell Signaling Technology), anti-p65 (IB:1:1000; #4764, Cell Signaling Technology), antibody to phosphorylated p65 (IB:1:1000; #3033, Cell Signaling Technology), anti-mTOR (IB:1:1000; #2983S; Cell Signaling Technology), anti-PI3 Kinase p85 (IB:1:1000; IP:1:100; #4257S; Cell Signaling Technology), anti-TBK1 (IF:1:500; Cat: NB100-56705, Novus Biologicals), antibody to phosphorylated PI3 Kinase p85 (IB:1:1000; #4228S, Cell Signaling Technology), anti-AKT1 (IB:1:1000; #2938S; Cell Signaling Technology), anti-AKT2 (IB:1:1000; #3063S; Cell Signaling Technology), anti-AKT3 (IB:1:1000; IP:1:100; #14982S; Cell Signaling Technology), antibody to phosphorylated AKT3 at Ser473 (IB:1:1000; 4060S, Cell Signaling Technology), anti-β-actin (IB:1:10000; A3854; Sigma), anti-HA (IB:1:5000; H6533; Sigma), anti-Myc (IB:1:5000; ab1326; Abcam), peroxidase affinipure goat anti-mouse light chain specific IgG (IB:1:10000; 115-035-174, Jackson ImmunoResearch), and peroxidase mouse anti-rabbit light chain specific IgG (IB:1:10000; 211-032-171, Jackson ImmunoResearch). The following antibodies were used for confocal microscopy assay: Alexa Fluor 488 goat anti-mouse secondary antibody (IF:1:1000; A-11001; ThermoFisher Scientific) and Alexa Fluor 594 goat anti-rabbit secondary antibody (IF:1:1000; R37117; ThermoFisher Scientific). Anti-HA and anti-Myc agarose beads were from Sigma. Lentiviral vectors for shRNA were from Dharmacon Inc. (Horizon Discovery Group company): PARP9 (clone TRCN0000052913 (PARP9-#1) and clone TRCN0000052915 (PARP9-#3)); PARP1 (clone TRCN0000007928); PARP2 (clone TRCN0000007933); PARP3 (clone TRCN0000052938); PARP4 (clone TRCN0000052923); PARP5 (clone TRCN0000040183); PARP6 (clone TRCN0000053203); PARP7 (clone TRCN0000004462); PARP8 (clone TRCN0000053223); PARP10 (clone TRCN0000052943); PARP11 (clone TRCN0000004535); PARP12 (clone TRCN0000004614); PARP13 (clone TRCN0000004516); PARP14 (clone TRCN0000053158); PARP15 (clone TRCN0000053003); PARP16 (clone TRCN0000053168); MAVS (clone TRCN0000146651); Akt1 (clone TRCN0000022936), Akt2 (clone TRCN0000055260), Akt3 (clone TRCN0000022836). Recombinant human GM-CSF (300-03), recombinant human IL-4 (200-04) and recombinant murine GM-CSF (315-03) were from PeproTech. Recombinant human IFN-α protein (11100-1) and recombinant mouse IFN-α protein (12100-1) were from R&D systems. Recombinant human PARP9 protein (ab79665) was from abcam. DNase I (M0303S), Rnase H (M0297S), Rnase A (T3018L), and Rnase III (M0245S) were from New England BioLabs. The IFN-β and IFN-α ELISA kits were from PBL Interferon Source. The TNF-α, IL-6 and IL-10 ELISA kits were from

R&D Systems. The Dual-Luciferase Reporter Assay System (E1910) was from Promega. Influenza A virus (PR8 A/Puerto Rico/8/1934(H1N1)) stocks, Reovirus (Reovirus type 3 strain Dearing, T3D), vesicular stomatitis virus (VSV, Indiana strain) and herpes simplex virus type I (HSV-1, KOS strain) were from ATCC (ATCC® VR-95™, ATCC® VR-824™, ATCC®VR-1238™, and ATCC®VR-1493™).

**Cell culture**. Human THP-1 cells (ATCC® TIB-202™) were differentiated to macrophages (THP-1 macrophages) with 60 nM phorbol 12-myristate 13-acetate (PMA; Sigma) for 16 h, and cells were cultured for an additional 48 h without PMA. Human monocyte-derived dendritic cells (MDDC) were isolated by stimulating isolated monocytes with granulocyte-macrophage colony-stimulating factor (GM-CSF, 1000 U/ml) and IL-4 (250 U/ml) and maintained in RPMI-1640 medium supplemented with 10% heat-inactivated FCS and 1% penicillin–streptomycin (Invitrogen-Gibco)[35,56,57]. Bone marrow cells were isolated from the tibia and femur and cultured in RPMI1640 medium with 10% FBS, 1% penicillin–streptomycin, and 10% L929 conditioned media containing macrophage-colony stimulating factor (M-CSF) for 6 days, 25 ng/ml murine GM-CSF for 6–8 days to harvest BMDM or BMDC, respectively[34,35]. For IFN-α treatment, human primary plasmacytoid dendritic cells (pDC) and myeloid dendritic cells (mDC) were isolated[35,58] and treated with IFN-α (100 U/ml) for 2 h. Mouse splenic CD11c + DCs were isolated from spleen of WT and PARP9 KO mice without or with VSV infection for 1 day using anti-CD11c microbeads (Miltenyi Biotec)[34].

**Lentivirus or retrovirus transduction and stimulation**. The pLKO.1 lentiviral vector carrying a scrambled shRNA or target gene sequences (Open Biosystems) were co-transfected into HEK 293FT cells (R70007; ThermoFisher Scientific) with packaging plasmids psPAX2 (Addgene 14858) and pMD2.G (Addgene 12259) using lipofectamine 3000 (ThermoFisher Scientific) for producing lentivirus. Human THP-1 macrophages and MDDC were infected by lentivirus as previously[59,60]. After 24 h of culture, cells were selected by the addition of puromycin (2 ng/ml) to the medium. The knockdown efficiency was detected with immunoblot analysis. The cDNA fragment encoding mouse PARP9 was amplified by PCR and further cloned into the pMYs-IRES-EGFP retroviral vector (Cell Biolabs). Retroviral particles were prepared by transfection of these vectors into packaging Plat-E cells according to the manufacturer's instructions (Cell Biolabs). The PARP9 KO or PARP9 and MAVS DKO BMDC or BMDM were transduced by incubation with freshly prepared retroviral particles by centrifugation for 2 h at 780 g and 30 °C in the presence of 10 μg/ml polybrene (Sigma-Aldrich)[61]. The cells after lentivirus or retrovirus transduction were stimulated for the indicated time with LPIC (0.5 μg/ml), SPIC (0.5 μg/ml) or 5′pppRNA (0.5 μg/ml) delivered by Lipofectamine 3000 or infected by Flu virus, reovirus or VSV at multiplicity of infection (MOI) of 5. The concentrations of IFN-α, IFN-β, TNF-α, and IL-6 in culture supernatants were measured by ELISA.

**In vivo virus infection**. For in vivo VSV infection study, age- and sex-matched Parp9[+/+]and Parp9[-/-] mice (n = 10 per strain, 6 weeks old) were infected with VSV ($5 \times 10^8$ PFU/mouse) by intraperitoneal injection. Additionally, age- and sex-matched PARP9[+/+] (WT) and PARP9[−/−] (KO) mice (n = 10 per strain, 6 weeks old) after intraperitoneal infection with VSV ($5 \times 10^8$ PFU per mouse) treated with 500 μg anti-IFNAR antibody (n = 10) or isotype control antibody (n = 10) starting 1 day before infection, followed by 250 μg three times per week for up to 2 weeks. Furthermore, age- and sex-matched PARP9[+/+] (WT), PARP9[−/−] (PARP9 KO), MAVS KO, and PARP9/MAVS double knockout (DKO) mice (n = 10 per strain, 6 weeks old) were infected with VSV ($1 \times 10^6$ PFU per mouse) by intraperitoneal infection. Serum cytokine production was measured by ELISA. The VSV titers in the lung, spleen, liver, and brain were determined by standard plaque assays. For the survival experiments, mice were monitored for survival after VSV infection.

For in vivo HSV-1 infection study, age- and sex-matched Parp9[+/+]and Parp9[-/-] mice were infected with HSV-1 ($1 \times 10^7$ PFU/mouse) by intravenous injection. Mice were monitored for survival after HSV-1 infection[35,62,63].

For reovirus infection in mice, age- and sex-matched Parp9[+/+]and Parp9[-/-] mice (n = 10 per strain, 6 weeks old) were inoculated intraperitoneally with $1 \times 10^7$ PFU of Reovirus (Reovirus type 3 strain Dearing, T3D) in 200 μL PBS. For the survival experiments, mice were monitored for survival after reovirus infection. Mice were euthanized at various time points following infection and tissues collected for analysis. For analysis of reovirus replication, mice were euthanized at defined intervals post-inoculation, and organs (heart, liver, spleen, brain, and intestine) were excised into PBS and homogenized by freezing, thawing, and sonication. Intestines were transected proximally at the gastroduodenal junction and distally at the rectum before homogenization in PBS.

**Virus titration**. After VSV infection in mice, total lung, spleen, and liver were removed and homogenized using PBS (pH 7.4). The supernatants from the homologous tissues were diluted and then used to infect confluent Vero cells (ATCC® CCL-81™) cultured on 12-well plates. At 1 h post-infection, the supernatant was removed, and 2% low melting-point agarose was overlaid. At 3-days post-infection, the overlay was removed and cells were fixed with methanol: acetic

acid solution (3:1 of methanol: acetic acid) for 20 min, and stained with 0.2% crystal violet. Plaques were counted, averaged, and multiplied by the dilution factor to determine viral titer as PFU/ml.

For reovirus infection in mice, viral titers in organ homogenates from infected mice were determined by plaque assay on L929 cells (ATCC® CCL-1™)[64]. Weights of organs were measured before the assay, and PFU was calculated per mg of tissue. Briefly, tissue was homogenized in 800 μl of PBS. The homogenates were treated with chloroform (10% final concentration), centrifuged briefly and serial dilutions of the aqueous supernatants were incubated on L929 cells at room temperature. After 1 h, the inoculum was removed and cells were covered with 2% agar solution with amphotericin-B. After 6 days, 2% agar solution containing 2% neutral red solution was added and plaques were visualized with neutral red on the second day.

**Histology**. Lungs were removed from naive or VSV infected wild-type and Parp9[-/-] mice, while liver and heart were removed from naive or reovirus infected wild-type and Parp9[-/-] mice. These removed tissues were washed using PBS before being fixed with 10% formaldehyde for 24 h at room temperature. The tissues were embedded in paraffin and processed by standard techniques. Longitudinal 5-μm sections were stained with Haematoxylin & Eosin (H&E)[65].

**RNA isolation, in vitro transcription, and biotin-label assay**. Viral genomic dsRNA from reovirus was isolated with QIAamp Viral RNA Mini Kit (Cat No: 52904, Qiagen) and were biotin-labeled with Pierce RNA 3′ End Biotinylation Kit (Cat No: 20160, ThermoFisher Scientific). The isolated reovirus genomic RNA was used to synthesize cDNA with the iScript cDNA Synthesis Kit (Bio-Rad). The viral cDNA was used as template to synthesize viral PCR fragments (8 to 1187 bp of reovirus S4 gene; 11 to 1198 bp of reovirus S3 gene; 16 to 1320 bp of reovirus S2 gene; 9 to 1410 bp of reovirus S1 gene) by PCR. The reovirus dsRNA including Reo 1187, Reo1198, Reo1320, and Reo1410 were synthesized using above PCR fragments with HiScribe T7 High Yield RNA Synthesis Kit (Cat No: E2040S, New England BioLabs) without 5′ppp or capped and were biotin-labeled with Pierce RNA 3′ End Biotinylation Kit.

**Pulldown assay**. To assay RNA binding to HA-tagged PARP9, PARP3, PARP13, PARP14 and PARP9 deletion mutants, HEK 293T cells (ATCC® CRL-3216™) were transfected with indicated plasmids. After 24 h, HEK 293T cells were washed twice, collected and lysed in buffer containing 50 mM Tris-HCL, 150 mM NaCl, 1% NP-40, 5 mM EDTA, protease inhibitor cocktail, and phosphatase inhibitor cocktail (ThermoFisher Scientific) for 30 min. Cell lysates were pelleted by centrifugation at 16,000 × g for 15 min at 4 °C and were incubated with anti-HA beads. Purified HA-tagged proteins were eluted from beads after six washings with PBS. The biotin-conjugated poly (I:C), poly(dA:dT), reovirus genomic dsRNA (ReoRNA), or Reo1198 were first immobilized onto 15 μL of streptavidin Sepharose beads. The beads were washed three times with lysis buffer (described above) to remove unconjugated ligands, and were then added to purified HA-tagged proteins or cell-free PARP9 recombinant protein to perform the standard pulldown assay. For the competition assay, the pre-cleared lysates were first incubated with 5, 10, or 20 μg of unlabeled poly(A), poly(I:C), viral dsRNA ReoRNA, or DNA poly(dA:dT) and CpG-B for 2 h at 4 °C with constant rotation, then added to biotinylated poly(I:C)-conjugated streptavidin beads to perform the standard pulldown assay.

**PARP9/Reo1198 RNA gel shift assay**. Aliquots of cell-free PARP9 recombinant protein (0.5, 1, 5 μg) were mixed with 1 pmol of biotinylated reovirus dsRNA Reo1198 (Reo1198) or unlabeled viral dsRNA Reo1198 (1, 5, 20 μg) for competitor assay, and incubated at room temperature for 1 h. Controls included, protein only and biotinylated-Reo1198 only reactions. After incubation for 1 h, proteins and biotinylated Reo1198 dsRNA were crosslinked with 1% formaldehyde for 30 min on ice. Crosslinking was quenched by adding glycine to a final concentration of 125 mM. Reactions were subjected to gel electrophoresis (Novex TBE-Urea 10% Gels, Invitrogen, Carlsbad, CA, USA) and transferred to nylon membranes (ThermoFisher Scientific). Biotin-labeled dsRNA was detected using the LightShift Chemiluminescent RNA EMSA Kit (ThermoFisher Scientific)[63,66].

**Co-immunoprecipitation of PARP9-bound RNA and RNAseq analysis**. WT and PARP9 KO BMDC ($10^7$ cells per 100 mm dish) were left infected or infected with reovirus at MOI of 200. Cells were harvested 16 h later and lysed in Nonidet P-40 (NP-40) lysis buffer (50 mM HEPES, pH 7.4, 150 mM KCl, 1 mM Na$_3$VO$_4$, 0.5% (v/v) NP-40 and 0.5 mM Dithiothreitol (DTT, Cat # 1019777001, Sigma), supplemented with protease inhibitor (ThermoFisher Scientific)) for 30 min at 4 °C. The lysates were cleared by centrifugation at 10,000 × g for 20 min at 4 °C. To co-immunoprecipitate PARP9-bound RNA, cleared lysates were mixed with anti-PARP9 or immunoglobulin G antibodies. After 3 h of incubation, the protein A/G beads were added for another 6 h incubation at 4 °C. Precipitates were washed three times with NP-40 lysis buffer and two times with high-salt wash buffer (50 mM HEPES, pH 7.4, 300 mM KCl, 1 mM Na$_3$VO$_4$, 0.5% (v/v) NP-40 and 0.5 mM DTT, supplemented with protease inhibitor (Sigma)), followed by incubation with proteinase K (New England Biolabs) at 55 °C for 1 h. The RNA bound to PARP9 was extracted using phenol/chloroform/isoamyl alcohol (Amresco).

RNAs purified from PARP9 precipitates from mock-infected or reovirus-infected WT or PARP9 KO BMDC (50 ng per sample) were utilized for generating RNA immunoprecipitaton (RIP) RNA library and RNA-seq using BGISEQ-500 platform (BGI Hong Kong, China). An average of 3.78 Gb bases were generated for each sample. Raw sequence reads were quality trimmed using TRIM Galore! (Cutadapt and FastQC wrapper) and were aligned to mammalian orthoreovirus 3 Dearing strain T3D (GenBank accession number HM 159622.1, HM 159621.1, HM 159620.1, HM 159619.1, HM 159618.1, HM 159617.1, HM 159616.1 HM 159615.1 HM 159614.1 HM 159613.1) or mouse genome (mm10) using Tophat2[67] with settings that allowed zero mismatches in the final sequencing alignment.

**Enzyme treatments of RNA**. About 0.5 g RNA isolated by RNA immunoprecipitation from WT BMDC left infected or infected with reovirus was treated with DNase I (0.2 U/μl), Rnase H (0.1 U/μl), Rnase A (1 μg/ml) with low (10 mM NaCl) or high (1 M NaCl) salt concentrations, and Rnase III (0.2 U/μl) at 37 for 1 h, followed by incubation with RNA loading dye. The RNA samples were then run on a 1% agarose gels in TBE (89 mM Tris-HCl pH 7.8, 89 mM Borate, 2 mM EDTA) and visualized on a ultra-violet (UV) transilluminator.

**In vitro co-immunoprecipitation and immunoblot analysis**. For the preparation of purified p85 and PARP9, HEK293T cells were transfected with expression plasmids encoding full-length or truncated versions of HA- or Myc-tagged p85 or PARP9. Lysates were prepared from the transfected cells, followed by incubation with anti-HA or anti-Myc beads. Proteins were eluted from the beads after beads were washed six times with PBS. For precipitation with anti-HA or anti-Myc beads, purified HA-tagged wild-type p85 or truncations of p85 were incubated for 2 h with purified Myc-tagged PARP9 or purified HA-tagged PARP9 or truncations of PARP9 were incubated for 2 h with purified Myc-tagged p85. Beads were added; after 2 h of incubation, the bound complexes were pelleted by centrifugation. Proteins and beads were analyzed by immunoblot analysis with anti-HA or anti-Myc Abs. For immunoprecipitation of endogenous proteins, the whole-cell lysates of WT and PARP9 KO BMDC left infected (Mock) or infected with VSV were incubated with anti-PARP9, anti-p85, anti-AKT3 or immunoglobulin G antibodies. After 2 h of incubation, the protein A/G beads were added for another 3 h incubation, and the bound complexes were pelleted by centrifugation. Proteins and beads were analyzed by immunoblot analysis with anti-PARP9, anti-mTOR, anti-p85, anti-AKT1, anti-AKT2, Anti-AKT3, anti-IRF3, and anti-IRF7 antibodies. WT, PARP9 KO, MAVS KO and PARP9/MAVS double knockout (DKO) BMDC were left infected (0 h) or infected with VSV for 2 h, 4 h or 6 h or reovirus for 4 h, and were then lysed in 1% NP-40 lysis buffer (50 mM Tris-HCl, 1%NP-40, 0.1% SDS, 150 mM NaCl) supplemented with protease inhibitor (ThermoFisher Scientific) followed by centrifugation. Supernatants were collected and incubated with SDS sample buffer by boiling of samples for 8 min followed by SDS-PAGE and immunoblot analysis. Immunoblot films were scanned by CanoScan 9000 F mark II and images were processed with Adobe Photoshop Creative Cloud (CC) 2019 (version 20.0.10).

**Dual-luciferase reporter assay**. HEK 293 T cells were transfected with the IFN-β luciferase reporter (IFN-β-Luc), TK-renilla, and HA-PARP9 plasmid or HA-vector by jetOPTIMUS DNA transfection reagent (Polyplus transfection) as per the manufacturer's instructions. At 20 h after transfection, cells were left stimulated or stimulated for 10 h with LPIC, or viral dsRNA including Reo1187, Reo1198, Reo1320, and Reo1410 (1 μg/ml) delivered by Lipofectamine 3000 transfection (ThermoFisher Scientific), reovirus, or influenza virus infection. After stimulation or reovirus infection, cells were lysed and then measured using a dual-luciferase reporter assay system (Promega) according to the manufacturer's instructions.

**Confocal microscopy**. The primary peritoneal macrophages were isolated from WT mice and then were mock or infected with VSV at MOI of 1 for 2 h. Next, the cells were fixed in 4% paraformaldehyde and permeabilized with 0.1% triton-100, then blocked for 1 h with 5% bovine serum albumin, incubated for 2 h with mouse anti-PI3 kinase p85 alpha monoclonal antibody (Cat: ab86714, Abcam), mouse anti-TBK1 monoclonal antibody (Cat: NB100-56705, Novus Biologicals) and rabbit anti-PARP9 polyclonal antibody (Cat: LS-B9440, LifeSpan BioSciences), followed by incubation for 1 h with Alexa Fluor 488 goat anti-mouse secondary antibody and Alexa Fluor 594 goat anti-rabbit secondary antibody and then examined with confocal microscopy[34,35]. Images of "zoomed" single cells were quantified with Nikon Confocal Software. Quantification of colocalization between PARP9 and p85 or TBK1 using ImageJ software[68].

**Quantitative RT-PCR**. RNA was isolated using the RNeasy Kit (Qiagen) according to the manufacturer's instructions. The isolated RNA was used to synthesize cDNA with the iScript cDNA Synthesis Kit (Bio-Rad). The quantitative RT-PCR (qRT-PCR) was performed on a CFX-96 real-time PCR detection system (Bio-Rad) with iTaq Universal SYBR Green Supermix (Bio-Rad). All qRT-PCR primers were listed in Supplementary Table S2.

**Flow cytometry**. Mouse spleen cells were isolated from *Parp9*+/+ (WT) and *Parp9*-/- (KO) mice infected with VSV for 1 day. The cells were then fixed and stained with Zombie Aqua fixable viability kit (423102, Biolegend), APC/Cyanine7 anti-mouse CD45 antibody (30-F11, Biolegend), FITC anti-mouse CD3 antibody (17A2, Biolegend), PE/Cyanine7 anti-mouse CD4 antibody (RM4-5, Biolegend), PerCP/Cyanine5.5 anti-mouse CD8a antibody (53-6.7, Biolegend) and APC anti-mouse CD19 antibody (1D3/CD19, Biolegend) (1 μl antibody for 2 million cells) for analyzing composition of CD4$^+$, CD8$^+$ and B cells. Flow cytometry data were acquired on a LSR-II flow cytometer (Beckton Dickinson) and analyzed using FlowJo v10 software (Tree Start)[34].

**Statistical analysis**. All samples sizes are large enough to ensure proper statistical analysis. Data are represented as the means ± DS of at least three experiments. Statistical analyses were performed using GraphPad Prism8 software (GraphPad Software, Inc.) and Microsoft Office Excel 2016. Statistical significance is calculated using Students's two tailed unpaired *t* test. The log-rank (Mantel–Cox) test is used for survival comparisons. NS, not significant ($p > 0.05$); *$p < 0.05$; **$p < 0.01$; ***$p < 0.001$.

**Reporting summary**. Further information on research design is available in the Nature Research Reporting Summary linked to this article.

## Data availability

All data generated or analyzed during the study are included in this published article and are available from the corresponding author upon reasonable request. The RNA sequencing (RNA-seq) data have been deposited with links to BioProject accession number "PRJNA609436" in the NCBI Sequence Read Archive (SRA) database. Source data are provided with this paper.

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

## Acknowledgements

We thank Dr. Sun Hur (Harvard University) for MAVS KO HEK 293T cells, Dr. Yuying Liang (University of Minnesota) for Vero cells, Dr. Lewis C. Cantley (Weill Cornell Medicine of Cornell University) for plasmid pCMV6-p85alpha-Flag, Dr. Lin Li (University of California Riverside) for plasmid pRK7-3×FLAG-PARP3, Dr. Mark R Boothby (University of Vanderbilt) for plasmid pcDNA3-FLAG/PARP14, and Dr. Kate Fitzgerald (University of Massachusetts Medical School) for plasmids pCMV-Flag-IRF3 and pCMV-Flag-IRF7. We also thank Beijing Genomics Institute (BGI) for their help with the RIP RNA-seq and bioinformatics analyses. Z.Z. is supported by Lupus Research Alliance grant 519418 and National Institutes of Health grant R56AI148215. X.C.L. is supported by the National Institutes of Health grant R01AI080779. J.X. is supported by the American Heart Association Career Development Award 20CDA35260116 (Xing). A.Z. is supported by the National Postdoctoral Program for Innovative Talents (BX20200399).

## Author contributions

J.X. designed and performed most of the experiments; A.Z., Y.D., M.F., and L.M. helped with some of the experiments; Z.Z. and Y.-J.L. supervised the project; J.X., X.C.L, and Z.Z. wrote the manuscript.

## Competing interests

The authors declare no competing interests.
