## [Peer Review File · Nature Communications]

REVIEWER COMMENTS

Reviewer #1 (Remarks to the Author):

Point 2. The new data in Fig 1l is not included in the revised manuscript.

Point 3a. It is highly surprising that anti-IFNAR treatment protects the wt and KO mice from virus-induced death. This does not at all fit with the hypothesis and conclusion of this work

Point 3b. The question raised on point 3 – as to whether PARP9 plays a role in the primary or positive feed-back loop of IFN induction – is not addressed through the data presented in Fig 3g.

Point 6. This is very interesting and strong data. However, I cannot find a description and call-out in the revised text.

Point 9. Based on the data presented in Fig 6e, the authors cannot conclude that there is not interaction/association between PARP9 and TBK1.

Point 12. The reproduction of data in human primary cells is not strong. Should preferentially be a functional data. E.g. demonstrating RNA-virus-induced PARP9-dependent IFN β induction.

Reviewer #2 (Remarks to the Author):

This paper is a revised version of a paper that had been submitted to Nature Cell Biology and reviewed by me. I was glad to see this paper had been revised and resubmitted for further consideration.

As I noted in my previous review, this paper addresses an interesting and timely topic. Overall, the study explores unique aspect of PARP9 biology and its mechanism of action in the innate immune system.

Strengths of the paper include a comprehensive analysis of PARP9 as a sensor of RNA virus. The use of the newly developed PARP9 knockout mice to study viral infections provides the necessary biological context for these studies. The combination of in vivo and cell-based studies with molecular and biochemical assays give the study a broader appeal. An interesting aspect of the study is the role of PARP9 as an activator of PI3K signaling cascade.

I also noted that the paper needed work to bring together the interesting observations in a more cohesive and less disjointed manner. Also, I noted that the role of PARP9 in antiviral responses has previously been shown in a number of studies, so the authors should further clarify how their proposed mechanism relates to those.

Overall, I think the reviewers have done a good job addressing my concerns. Their extensive edits and new data have improved the paper considerably. All of my concerns have been addressed.

Reviewer #3 (Remarks to the Author):

This is a revised manuscript. The authors have adequately responded to previous comments through a combination of text revisions and performing additional experiments. The presentation is improved such that the data shown does support the conclusions that PARP9 is a sensor of dsRNA and operates likely downstream of RLR and maybe TLR signaling to amplify IFN defenses in specific cell types including dendritic cells. The data is overall interesting to reveal PARP9 pathway as polarized to IRF3 activation rather than to NF- κ B activation and is hence more antiviral function than inflammatory function. This point really could be discussed in more depth in the discussion, so I do suggest that the authors should consider including a specific paragraph in the discussion to make the point that PARP9 signaling directs IRF3/IFN antiviral response in a manner that does not highly engage the inflammatory signaling cascades directed by NF- κ B. This aspect of the observations make the study increasingly novel, and present a scenario in which one could consider therapeutic targeting of PARP9 signaling to treat virus infection and avoid enhancing a inflammatory cytokine storm. I thank the authors for addressing my comments.

Point-by-point reply

We thank the reviewers for their constructive suggestions and comments. In this revised version we have performed additional experiments, added new data, and revised the manuscript accordingly. We believe that the revised manuscript has addressed all of the concerns, and with these revisions the quality of the manuscript is also markedly improved. All changes in the revised manuscript are highlighted in yellow for your attention. The following is a point by point reply on how we revised this manuscript.

REVIEWER COMMENTS

Reviewer #1 (Remarks to the Author):

Point 2. The new data in Fig 1l is not included in the revised manuscript.

Reply: We appreciate the reviewer's comment. It's our mistake to put Fig. 1l right under Fig. 1g and make Fig. 1l be not easily visible previously. We now reorganize the figures and put on Fig. 1l right above the Fig. 1m so that Fig. 1l is easily visible.

Point 3a. It is highly surprising that anti-IFNAR treatment protects the wt and KO mice from virus-induced death. This does not at all fit with the hypothesis and conclusion of this work

Reply: Excellent points. Anti-IFNAR treatment enhances mice death after infection by virus. It's our mistakes to mix the raw data and put the wrong figure and data in Fig. 3g, which was noticed by us when we have submitted the manuscript and could not correct it. We have now put on the right figure and data in Fig. 3g in revised manuscript.

Point 3b. The question raised on point 3 – as to whether PARP9 plays a role in the primary or positive feed-back loop of IFN induction – is not addressed through the data presented in Fig 3g.

Reply: Great suggestions. We now have isolated CD11c+ splenocytes from WT and PARP9 KO mice treated with isotype control or anti-IFNAR antibody without or with VSV infection and detected PARP9 expression in Fig. 3h. PARP9 was dramatically induced by IFN with VSV infection, while PARP9 was significantly reduced due to IFNAR blockage even if VSV infection in WT mice. Additionally, our data in Fig. 1g and 1l also showed that PARP9 expression was strongly induced by IFN- α in human DCs. Collectively, PARP9 plays a role in positive feed-back loop of IFN induction.

Point 6. This is very interesting and strong data. However, I cannot find a description and call-out in the revised text.

Reply: We thank the reviewer for your good suggestion. It's our mistake not to highlight and call out the description and make it easily visible in the text. We have highlighted and called out the description about Supplementary Fig. 12 in lines 329-334 in page 14.

Point 9. Based on the data presented in Fig 6e, the authors cannot conclude that there is not

interaction/association between PARP9 and TBK1.

Reply: We agree with the reviewer's comment. The data in Fig. 6e only showed there was no colocalization between PARP9 and TBK1, however, it could not demonstrate there was not interaction of PARP9 and TBK1. Our IP-MS data in supplementary Table 1 showed that TBK1 is not in the PARP9-binding protein complex. Based on the data in Fig. 6e and IP-MS data, we believe no interaction between PARP9 and TBK1.

Point 12. The reproduction of data in human primary cells is not strong. Should preferentially be a functional data. E.g. demonstrating RNA-virus-induced PARP9-dependent IFN β induction.

Reply: This is an excellent question. Given that PARP9 was very lowly expressed and induced by IFN- α in human monocyte-derived dendritic cells (MDDC), we performed experiments to treat firstly human MDDC with or without MAVS or PARP9 knockdown using IFN- α for 2h and then stimulated those cells with intracellular LPIC or VSV infection for IFN- β detection by ELISA. Knockdown of MAVS or both MAVS and PARP9 abrogated almost the production of IFN- β compared to the control human MDDC in response to intracellular LPIC or VSV infection (Supplementary Fig. 13). The MAVS knockdown human MDDC with IFN- α treatment produced more IFN- β than those cells without IFN- α treatment (Supplementary Fig. 13), suggesting PARP9 indeed initiates and amplifies MAVS-independent induction of type I IFN in human MDDC.

Reviewer #2 (Remarks to the Author):

This paper is a revised version of a paper that had been submitted to Nature Cell Biology and reviewed by me. I was glad to see this paper had been revised and resubmitted for further consideration.

As I noted in my previous review, this paper addresses an interesting and timely topic. Overall, the study explores unique aspect of PARP9 biology and its mechanism of action in the innate immune system.

Strengths of the paper include a comprehensive analysis of PARP9 as a sensor of RNA virus. The use of the newly developed PARP9 knockout mice to study viral infections provides the necessary biological context for these studies. The combination of in vivo and cell-based studies with molecular and biochemical assays give the study a broader appeal. An interesting aspect of the study is the role of PARP9 as an activator of PI3K signaling cascade.

I also noted that the paper needed work to bring together the interesting observations in a more cohesive and less disjointed manner. Also, I noted that the role of PARP9 in antiviral responses has previously been shown in a number of studies, so the authors should further clarify how their proposed mechanism relates to those.

Overall, I think the reviewers have done a good job addressing my concerns. Their extensive edits and new data have improved the paper considerably. All of my concerns have been addressed.

Reply: We appreciate the reviewer 2's very positive comments and full support.

Reviewer #3 (Remarks to the Author):

This is a revised manuscript. The authors have adequately responded to previous comments through a combination of text revisions and performing additional experiments. The presentation is improved such that the data shown does support the conclusions that PARP9 is a sensor of dsRNA and operates likely downstream of RLR and maybe TLR signaling to amplify IFN defenses in specific cell types including dendritic cells. The data is overall interesting to reveal PARP9 pathway as polarized to IRF3 activation rather than to NF- κ B activation and is hence more antiviral function than inflammatory function. This point really could be discussed in more depth in the discussion, so I do suggest that the authors should consider including a specific paragraph in the discussion to make the point that PARP9 signaling directs IRF3/IFN antiviral response in a manner that does not highly engage the inflammatory signaling cascades directed by NF- κ B. This aspect of the observations make the study increasingly novel, and present a scenario in which one could consider therapeutic targeting of PARP9 signaling to treat virus infection and avoid enhancing a inflammatory cytokine storm. I thank the authors for addressing my comments.

Reply: We thank reviewer 3 for her/his very positive comments. We have now included description that PARP9 signaling directs IRF3/IFN antiviral response in a manner that does not highly engage the inflammatory signaling cascades directed by NF- κ B in the discussion section, which is highlighted in yellow.

REVIEWER COMMENTS

Reviewer #1 (Remarks to the Author):

I find that the authors have now complied with the points I have raised in a satisfactory manner, and I would like to see the work in published form.

Point-by-point reply

REVIEWERS' COMMENTS

Reviewer #1 (Remarks to the Author):

I find that the authors have now complied with the points I have raised in a satisfactory manner, and I would like to see the work in published form.

Reply: We thank the reviewer for the final acceptance of our manuscript.